# Tumor suppressor Par-4 activates autophagy-dependent ferroptosis
Karthikeyan Subburayan[1], Faisal Thayyullathil[1], Siraj Pallichankandy[1], Anees Rahman Cheratta[1], Ameer Alakkal[1], Mehar Sultana[2], Nizar Drou[3], Muhammad Arshad[3], L. Palanikumar [4], Mazin Magzoub [4], Vivek M. Rangnekar[5] & Sehamuddin Galadari [1] ✉

Ferroptosis is a unique iron-dependent form of non-apoptotic cell death characterized by devastating lipid peroxidation. Whilst growing evidence suggests that ferroptosis is a type of autophagy-dependent cell death, the underlying molecular mechanisms regulating ferroptosis are largely unknown. In this study, through an unbiased RNA-sequencing screening, we demonstrate the activation of a multi-faceted tumor-suppressor protein Par-4/PAWR during ferroptosis. Functional studies reveal that genetic depletion of Par-4 effectively blocks ferroptosis, whereas Par-4 overexpression sensitizes cells to undergo ferroptosis. More importantly, we have determined that Par-4-triggered ferroptosis is mechanistically driven by the autophagic machinery. Upregulation of Par-4 promotes activation of ferritinophagy (autophagic degradation of ferritin) via the nuclear receptor co-activator 4 (NCOA4), resulting in excessive release of free labile iron and, hence, enhanced lipid peroxidation and ferroptosis. Inhibition of Par-4 dramatically suppresses the NCOA4-mediated ferritinophagy signaling axis. Our results also establish that Par-4 activation positively correlates with reactive oxygen species (ROS) production, which is critical for ferritinophagy-mediated ferroptosis. Furthermore, Par-4 knockdown effectively blocked ferroptosis-mediated tumor suppression in the mouse xenograft models. Collectively, these findings reveal that Par-4 has a crucial role in ferroptosis, which could be further exploited for cancer therapy.

Ferroptosis is a non-apoptotic regulated cell death pathway driven by iron-mediated production of reactive oxygen species (ROS) and subsequent lipid peroxidation[1,2]. Increasingly, studies indicate that ferroptosis is not a self-standing phenomenon, and that it has close connections with other cellular events[3]. Recent insights demonstrate that ferroptosis is an autophagy-dependent cell death, and a highly conserved lysosomal degradation pathway, responsible for recycling dysfunctional cellular components under survival stress[4–6]. Autophagic machinery, including the nuclear receptor co-activator 4 (NCOA4)-mediated ferritinophagy (autophagic degradation of key intracellular iron-storage protein ferritin)[7], p62/SQSTM1-mediated clockophagy (autophagic degradation of the circadian clock regulator ARNTL)[8], RAB7A-dependent lipophagy (autophagic degradation of lipid droplets)[9], chaperone-mediated autophagic degradation of glutathione

peroxidase 4 (GPX4)[10], beclin1-mediated system Xc⁻ inhibition[11], and signal transducer and activator of transcription 3 (STAT3)-induced lysosomal membrane permeabilization can significantly contribute to ferroptosis[12]. The vulnerability to ferroptosis is closely associated with numerous biochemical processes, including (i) amino acid, iron, and polyunsaturated fatty acid (PUFAs) metabolism, and (ii) biosynthesis of glutathione (GSH), phospholipids (arachidonic acid, and adrenic acid), nicotinamide adenine dinucleotide phosphate (NADPH), and coenzyme Q10[2]. Moreover, small-molecule ferroptosis inducers (FINs), such as eradicator of Ras and Small T oncoprotein (Erastin), and 1 S,3R-Ras Selective Lethal 3 (RSL3), have been well established as pharmacological activators of ferroptosis[13]. These FINs either inhibit the cystine/glutamate antiporter (system Xc⁻) or phospholipid GPX4, leading to decreased antioxidant capacity and uncontrolled

[1]Cell Death Signaling Laboratory, Division of Science (Biology), Experimental Research Building, New York University Abu Dhabi, PO Box 129188 Saadiyat Island, Abu Dhabi, United Arab Emirates. [2]Center for Genomics and Systems Biology (CGSB), Experimental Research Building, New York University Abu Dhabi, PO Box 129188 Saadiyat Island, Abu Dhabi, United Arab Emirates. [3]CGSB Core Bioinformatics, Experimental Research Building, New York University Abu Dhabi, PO Box 129188 Saadiyat Island, Abu Dhabi, United Arab Emirates. [4]Biology Program, Division of Science, Experimental Research Building, New York University Abu Dhabi, PO Box 129188 Saadiyat Island, Abu Dhabi, United Arab Emirates. [5]Department of Radiation Medicine and Markey Cancer Center, University of Kentucky, Lexington, KY 40536, USA. ✉e-mail: sehamuddin@nyu.edu

accumulation of lipid peroxides, eventually causing the cell to die[1,2]. Emerging evidence shows that ferroptosis plays an essential role in different pathological processes, including cancer, ischemic tissue injuries (e.g., ischemic heart diseases, brain damage, kidney failure, and lung injury), and neurodegeneration (e.g., Parkinson's, Huntington's and Alzheimer's diseases)[2,13]. For instance, knockout of GPX4, a vital protein involved in ferroptosis regulation, in mice leads to ferroptosis and is associated with acute renal failure[14]. In addition, intracerebral hemorrhage-induced ferroptosis contributes to neuronal death in mouse models[15]. Thus, an improved understanding of the process and function of ferroptosis will yield cutting-edge opportunities for diagnosing and treating cancer and other diseases, including neurodegenerative disorders and ischemic tissue injuries[1].

Prostate apoptosis response-4 (Par-4), encoded by the PAWR gene (also named PRKC apoptosis WT1 regulator), is a therapeutically promising tumor-suppressor protein that can selectively induce apoptosis in cancer cells[16-18]. Moreover, Par-4 is ubiquitously expressed in almost all cell types across different species[19]. Importantly, Par-4 is depleted, mutated, or inactivated in multiple cancers[20-24]. For instance, loss or mutation of genetic material around chromosome 12q21, which harbors the human PAWR gene, is noted in Wilms' tumorigenesis[24,25] and human male germ cell tumor development[26]. Indeed, Par-4 knockout mice spontaneously develop tumors in various organs, while transgenic mice overexpressing Par-4 are resistant to the growth of spontaneous or oncogene-inducible tumors[27,28]. The importance of Par-4-mediated apoptosis in its tumor-suppressive functions has been deciphered in depth. In fact, previous studies, including work from our laboratory, demonstrated that Par-4 could also activate other tumor-suppressive mechanisms such as autophagy[29,30], senescence[31], and anti-metastasis[32]. However, the role and mechanism of Par-4 in inducing ferroptosis have not been previously investigated. In this study, we report that Par-4 plays a hitherto unanticipated role in promoting ferroptosis both in vitro and in vivo in glioblastoma (GBM) cells. Mechanistically, we demonstrate that Par-4 induces the activation of ferritinophagy via NCOA4, which is required for accumulation of the labile iron pool, ROS production, and subsequent lipid peroxidation leading to ferroptosis. Thus, Par-4 contributes to the core molecular machinery and signaling pathways involved in ferroptosis.

## Results

### Identification of Par-4 as a pro-ferroptosis gene

Changes in molecular and metabolic basis that underlies the regulation of ferroptosis are largely elusive. Therefore, we screened unbiased global gene expression to identify unique factors modulating ferroptosis vulnerability. To this end, we treated U87MG cells with type 2 ferroptosis activator RSL3 and conducted RNA sequencing (RNA-seq) analysis. RNA-seq identified 1232 differentially expressed genes (DEGs) (638 upregulated and 594 downregulated genes) between the control and RSL3-treated groups, as shown in the volcano plot ($P < 0.05$) Fig. 1a and Supplementary Data 1. The known ferroptosis regulators, including activating transcription factor 3 (ATF3)[33], cytochrome P450 oxidoreductase (POR)[34], kelch-like ECH associated protein 1 (KEAP1)[35], GPX4[36], NCOA4[37], mitogen-activated protein kinase 1 (MAPK1)[38], acyl-CoA synthetase long-chain family member 4 (ACSL4)[39], and heme oxygenase 1 (HMOX1)[40], were differentially identified in RSL3-treated U87MG cells, confirming the robustness of our screening (Fig. 1a, b and Supplementary Data 1). Additionally, we identified several uncharacterized pro-ferroptosis genes in our analysis (Supplementary Data 1). Notably, in RSL3-treated U87MG cells, PAWR (Par-4) was significantly upregulated during ferroptosis (Fig. 1a, b, and Supplementary Data 1), suggesting a role of Par-4 in regulating ferroptosis. To verify the results of the RNA-seq screening, we examined the mRNA expression of the Par-4 gene in these groups through qRT-PCR. We observed a time-dependent increase in the relative mRNA expression level of Par-4 in RSL3-treated cells (Fig. 1c). The RSL3-induced Par-4 expression in U87MG cells in a dose- and time-dependent manner was further confirmed with Western blot analysis (Fig. 1d). Similar results were also found

in other glioma cell lines (A172) (Fig. 1e). To ensure the accuracy and reliability of these screenings, we used another well-established type 1 classical ferroptosis activator, erastin. Consistent with RSL3, erastin treatment also significantly upregulated the expression of Par-4 at both mRNA and protein levels in A172 cells (Fig. 1f, g). Interestingly, the increase in Par-4 expression caused by RSL3 was hindered by potent lipid peroxidation scavengers ferrostatin-1 (Fer-1) and liproxstatin-1 (Lip-1), indicating that the response of Par-4 activation was partly due to lipid peroxidation accumulation linked to ferroptosis (Fig. 1h). Additionally, we tested the activation of Par-4 in ferroptosis by using a direct GPX4 (a crucial defense enzyme for lipid peroxidation and ferroptosis) inhibitor, ML210. We observed a dose-dependent upregulation of Par-4 upon treatment with ML210 (Supplementary Fig. 1a). Together, these results demonstrate that Par-4 is activated during ferroptosis, irrespective of the specific glioma cell types and ferroptosis inducers.

Next, we checked the anticancer activity of RSL3- and erastin in U87MG and A172 cells. As shown in Supplementary Fig. 1b, c, RSL3 and erastin cause dose-dependent cell death. Given that glioma cell death also occurs via apoptosis and other types of regulated cell death, we explored the possibility that this regulated cell death pathway might contribute to the cytotoxic effects of erastin and RSL3. Interestingly, we observed that the growth inhibition induced by RSL3 and erastin was not affected by apoptosis or necroptosis inhibitors (Supplementary Fig. 1d–f) but was blocked by lipid peroxidation scavengers (Fer-1 and Lip-1) and iron chelator deferoxamine (DFO) (Supplementary Fig. 1g–i).

### Par-4 activation is critical for ferroptosis in human glioblastoma cells

To explore the pro-ferroptosis role of Par-4, we probed whether Par-4 inhibition by specific short hairpin RNA (shRNA) affects ferroptosis in GBM cells. The suppression of Par-4 significantly inhibited both RSL3- and erastin-induced GPX4 protein degradation (Fig. 2a). Subsequently, we determined whether Par-4 knockdown affects the GPX4 activity. The suppression of Par-4 significantly increased GPX4 activity in erastin-treated A172 cells. Interestingly, Par-4 knockdown alone had higher GPX4 activity than the control shRNA group (Supplementary Fig. 2a). Moreover, Par-4 inhibition also prevented labile iron accumulation (Fig. 2b and Supplementary Fig. 2b), and lipid peroxidation (Fig. 2c), which are surrogate markers for ferroptosis. Additionally, suppression of Par-4 significantly prevented both RSL3- and erastin-induced cell death (Fig. 2d). A CRISPR/Cas9-mediated genome editing tool was used to corroborate these results by Par-4 knockout in U87MG cells. CRISPR/Cas9-mediated knockout of Par-4 (Supplementary Fig. 2c) significantly inhibited RSL3-induced GPX4 degradation (Fig. 2e). Similar results were obtained using a small interfering RNA (siRNA)-mediated knockdown of Par-4 in U87MG and A172 cells (Supplementary Fig. 2d–g). To strengthen the functional importance of Par-4 in ferroptosis indicated by our results so far, we used Par-4$^{+/+}$ and Par-4$^{-/-}$ primary mouse embryonic fibroblasts (MEFs). Consistently, homozygous knockout Par-4$^{-/-}$ MEFs dramatically suppressed RSL3, ML210-and erastin-induced GPX4 degradation (Fig. 2f and Supplementary Fig. 2h), labile iron accumulation (Fig. 2g and Supplementary Fig. 2i–k), lipid peroxidation (Fig. 2h), and ferroptosis (Fig. 2i, j and Supplementary Fig. 2l). Notably, the colony-forming efficiency was significantly reduced by RSL3 in Par-4$^{+/+}$ compared to Par-4$^{-/-}$ MEFs (Fig. 2k), lending further support for the suggested critical role of Par-4 activation during ferroptosis. It should be noted that Par-4 knockout or knockdown alone was sufficient to protect the basal GPX4 degradation (Fig. 2a, e, f, and Supplementary Fig. 2d), implicating a direct role of Par-4 in the execution of ferroptosis. To further investigate the pro-ferroptosis role of Par-4, we examined whether Par-4 overexpression potentiates RSL3-induced ferroptosis in U87MG and A172 cells. Transient-enforced overexpression of Par-4 resulted in significant sensitization of RSL3-induced GPX4 degradation, labile iron accumulation, lipid peroxidation, and ferroptosis in both U87MG and A172 cells (Fig. 2l–p). Surprisingly, overexpression of Par-4 alone was sufficient to induce most of these features of ferroptosis (Fig. 2l–p). To further confirm

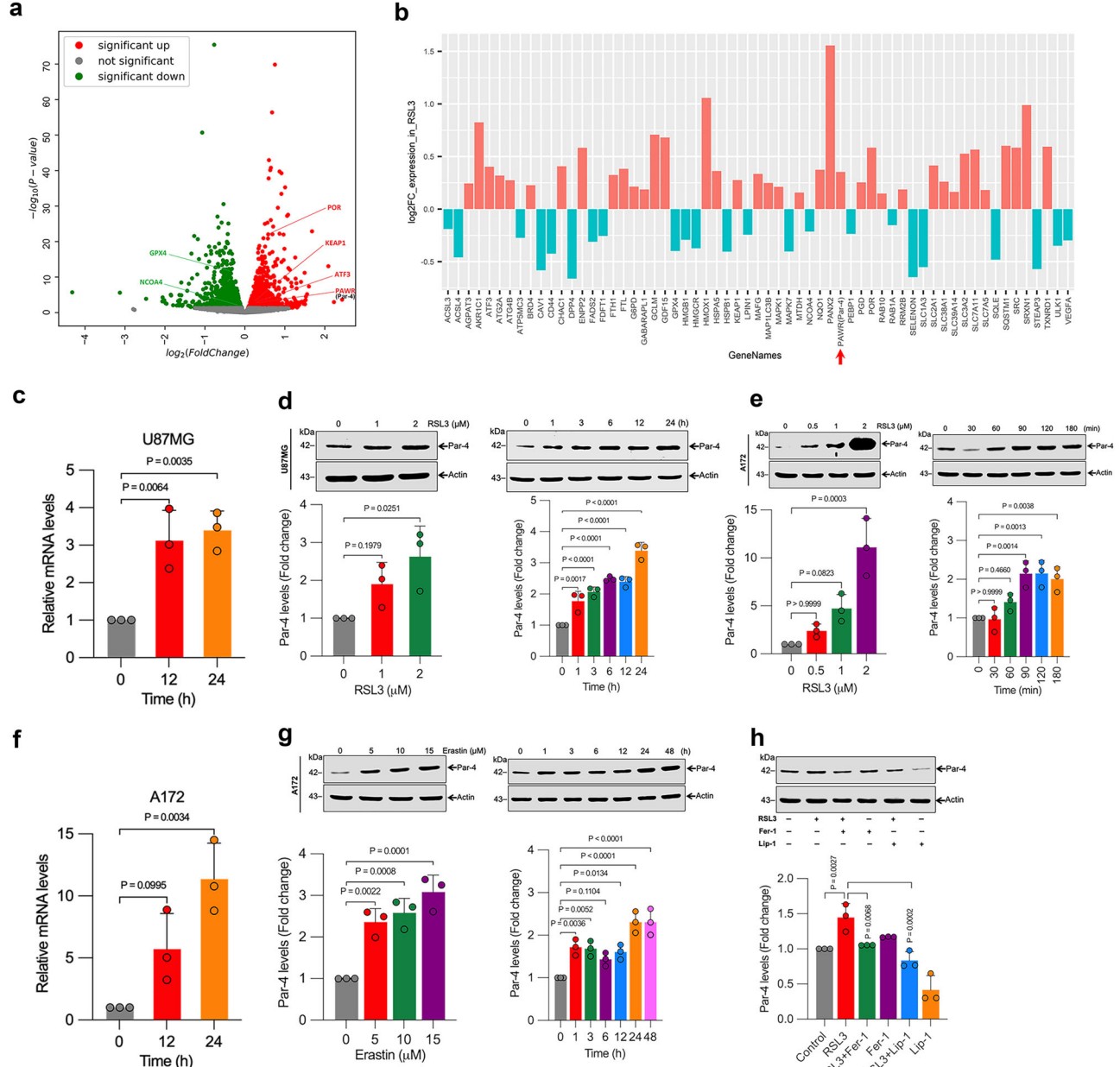

**Fig. 1 | Identification of PAWR/Par-4 as a critical contributor of ferroptosis.** U87MG cells were treated with RSL3 (2 μM) for 24 h. Total RNA was isolated for subjected to RNA-Seq analysis and identified PAWR (Par-4) along with ferroptosis-related differentially expressed genes (DEGs). **a** Volcano plot showing the DEGs of gene expression where the log2-fold change of mRNA signal was plotted against the negative log10-transformed P values for differential mRNA abundance; *n* = 3 samples. Exact *P* values are given in Supplementary Data 1. The arrows are labeled with the gene symbols. Red highlights upregulated genes, green highlights downregulated genes, and gray dots highlight non-significant genes. **b** The bar plot demonstrates the log2-fold change in expression of the ferroptosis-related genes following RSL3 treatment compared with control U87MG cells (*n* = 3 samples). The selected genes were identified as related to ferroptosis using FerrDb at http://www.zhounan.org/ferrdb/. The bars are labeled with gene symbols. Significantly upregulated and downregulated genes are depicted in red and green color, respectively. The gene of interest, PAWR (Par-4), is highlighted by the red arrow. **c** Indicated cells were treated with RSL3 (2 μM) for the indicated time period, and relative Par-4 mRNA levels were assessed by qRT-PCR. Data shown are mean ± SD; *n* = 3 samples. **d** U87MG and **e** A172 cells were treated with the indicated concentration of RSL3 for

3 h and 2 μM RSL3 for the indicated time period, and then Western blot analysis of Par-4 was determined. The relative density of protein bands were quantified and normalized to the actin of each group, and fold changes were presented in histograms from three independent experiments. **f** Indicated cells were treated with erastin (10 μM) for the indicated time period, and Par-4 mRNA levels were assessed by qRT-PCR. Data shown are mean ± SD; *n* = 3 samples. **g** A172 cells were treated with the indicated concentration of erastin for 24 h and 10 μM erastin for the indicated time period, and then Western blot analysis of Par-4 was determined. The relative density of protein bands were quantified and normalized to the actin of each group, and fold changes were presented in histograms from three independent experiments. **h** A172 cells were treated with RSL3 (2 μM) for 3 h in the presence or absence of Fer-1 (5 μM) and Lip-1 (1 μM). Following the treatment, Western blot analysis of Par-4 was detected. The relative density of protein bands were quantified and normalized to the actin of each group, and fold changes were presented in histograms from three independent experiments. Data shown are mean ± SD; *n* = 3 samples. Statistical significance (*P* values) was analyzed by one-way ANOVA using the Bonferroni post-hoc test.

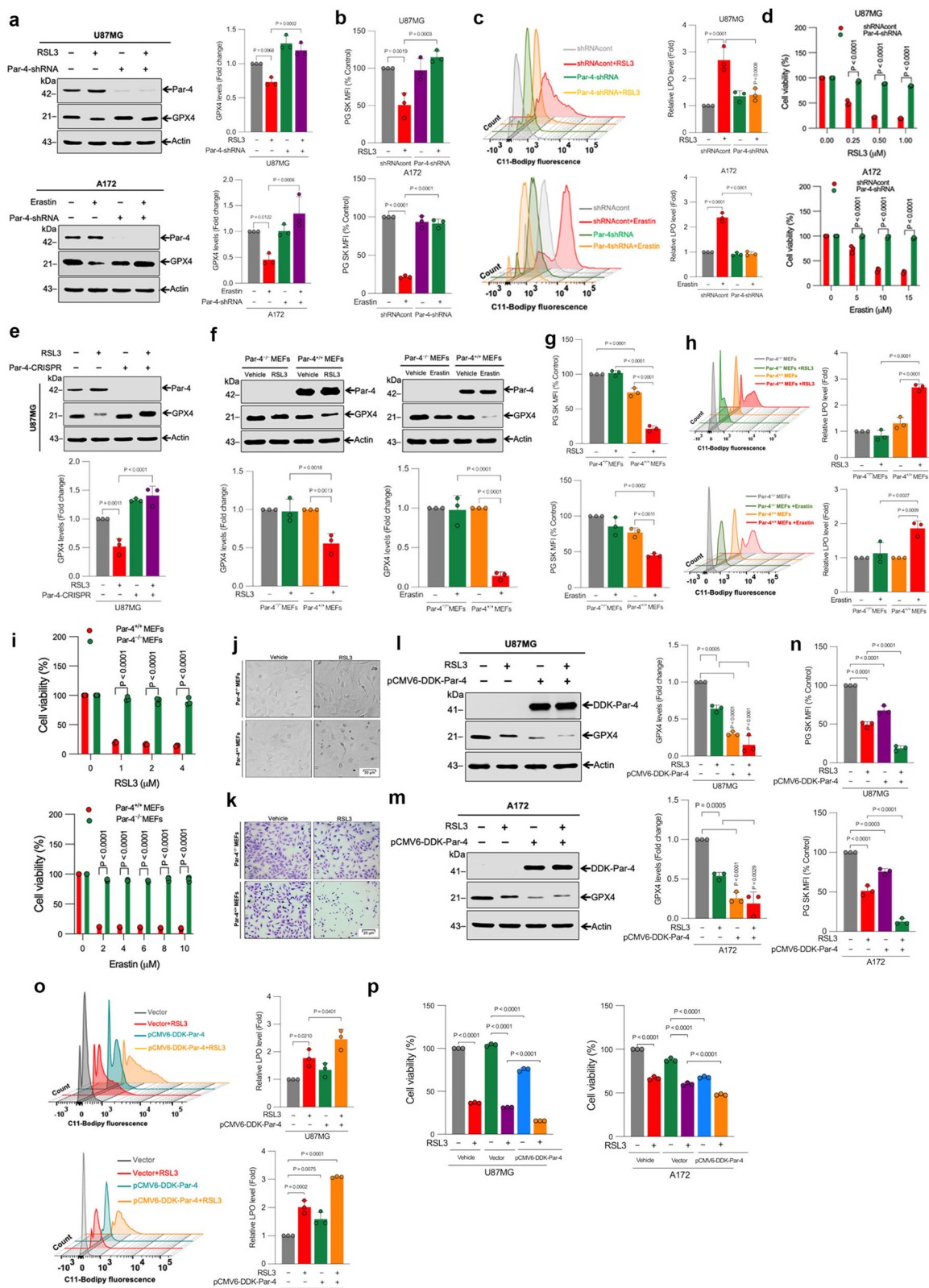

the involvement of Par-4 in ferroptosis, we generated doxycycline-controlled (Dox-ON) Par-4 inducible U87MG cells. Upon Par-4 expression induced by doxycycline, degradation of GPX4 and ferroptosis was dramatically increased following RSL3 treatment (Supplementary Fig. 2m, n). Collectively, these findings prove that Par-4 plays a crucial role in regulating ferroptosis.

p53 is a master regulator of tumorigenesis, and studies have indicated that p53 is a positive regulator of ferroptosis[41,42]. Mechanistically, p53 was reported to sensitize cells to ferroptosis through transcriptionally repressing the expression of solute carrier family 7 member 11 (SLC7A11), a member of a heteromeric anionic cystine/glutamate antiporter[41]. Previously, we and others have shown that Par-4 induces tumoricidal function via activating

**Fig. 2 | Par-4 activation is essential for ferroptosis. a** Indicated cells were stably transfected with Par-4-shRNA. After transfection, cells were treated with RSL3 (2 µM) for 3 h and erastin (10 µM) for 24 h, and then Western blot analysis of Par-4 and GPX4 were carried out. The relative density of protein bands were quantified and normalized to the actin of each group, and fold changes were presented in histograms from three independent experiments. **b** Indicated cells were stably transfected with Par-4-shRNA. After transfection, cells were treated with RSL3 (2 µM) for 3 h and erastin (10 µM) for 24 h. Following the treatment, intracellular labile iron was determined using flow cytometry. The bar graph showing labile iron level was expressed as a percentage of the control. Data shown are mean ± SD; $n = 3$ samples. **c** Indicated cells were stably transfected with Par-4-shRNA. After transfection, cells were treated with RSL3 (2 µM) for 3 h and erastin (10 µM) for 24 h. Following the treatment, cells were stained with a C11-Bodipy 581/591 fluorescence probe, and lipid peroxidation was detected by flow cytometry. Bar graph showing the relative levels of lipid peroxidation. Data shown are mean ± SD; $n = 3$ samples. **d** Indicated cells were stably transfected with Par-4-shRNA. After transfection, cells were treated with the indicated concentration of RSL3 and erastin for 24 h. Cell viability was measured following the treatment using an MTT assay. Data shown are mean ± SD; $n = 3$ samples. **e** U87MG cells were stably transfected with the Par-4-CRISPR-Cas9 system. After transfection, cells were treated with RSL3 (2 µM) for 3 h, and then Western blot analysis of Par-4 and GPX4 were carried out. The relative density of protein bands were quantified and normalized to the actin of each group, and fold changes were presented in histograms from three independent experiments. Par-4$^{+/+}$ and Par-4$^{-/-}$ MEFs were treated with RSL3 (2 µM) for 3 h and erastin

(10 µM) for 24 h. Following the treatment, **f** Western blot analysis of indicated proteins were carried out. The relative density of protein bands were quantified and normalized to the actin of each group, and fold changes were presented in histograms from three independent experiments. **g** Intracellular labile iron was determined using flow cytometry. The bar graph showing labile iron level was expressed as a percentage of the control. **h** Lipid peroxidation was detected by flow cytometry. Bar graphs showing the relative lipid peroxidation levels and **i** Cell viability by MTT assay were performed. Data shown are mean ± SD; $n = 3$ samples. Par-4$^{+/+}$ and Par-4$^{-/-}$ MEFs were treated with RSL3 (2 µM) for 3 h. Following treatment, **j** Representative images of phase contrast and **k** Crystal violet staining were performed. Data shown are mean ± SD; $n = 3$ samples. **l** U87MG and **m** A172 cells were transiently transfected with pCMV6-DDK-Par-4. After transfection, cells were treated with RSL3 (2 µM) for 3 h. Following the treatment, a Western blot analysis of indicated proteins were carried out. The relative density of protein bands were quantified and normalized to the actin of each group, and fold changes were presented in histograms from three independent experiments. U87MG and A172 cells were transiently transfected with pCMV6-DDK-Par-4. After transfection, cells were treated with RSL3 (2 µM) for 3 h. Following the treatment, **n** Intracellular labile iron was determined using flow cytometry. The bar graph showing labile iron level was expressed as a percentage of the control. **o** Lipid peroxidation was detected by flow cytometry. Bar graphs showing the relative lipid peroxidation levels and **p** Cell viability using an MTT assay were determined. Data shown are mean ± SD; $n = 3$ samples. Statistical significance (*P* values) was analyzed by one-way ANOVA using the Bonferroni post-hoc test.

---

p53[30,31,43]. Therefore, in the current study, we wanted to ascertain whether p53 has any role in RSL3-induced ferroptosis. Western blot analysis showed that RSL3 decreased the expression of p53 over time in U87MG cells (Supplementary Fig. 2o), suggesting that Par-4-mediated ferroptosis induced by RSL3 is independent of p53 expression.

## Par-4 regulates ferroptosis through autophagy

Several lines of evidence, including recent work from our lab, proposed that ferroptosis is a type of autophagy-dependent cell death[5–7,44]. However, the mechanisms that govern specific forms of autophagy-dependent ferroptosis have not been characterized in GBM cells. Therefore, we next wanted to validate if autophagy is, indeed, responsible for the RSL3- and erastin-induced ferroptosis in glioma cells. To address this question, we first examined the target genes involved during autophagy favoring ferroptosis signaling. Interestingly, some autophagy-related gene profiles, such as MAP1LC3B, p62/SQSTM1, ULK1, GABARAPL1, ATG2A, ATG4B, RAB1A, and RAB10, were differentially expressed in RSL3-treated U87MG cells (Fig. 1b and Supplementary Data 1), and these positive outcomes indicated that autophagic machinery may be involved in the facilitation of ferroptosis. Microtubule-associated protein light chain 3-II (LC3-II) conversion, p62/SQSTM1 degradation, and autolysosome formation are three well-documented hallmarks of autophagy linked with ferroptosis[3,45]. Subsequently, our Western blot analysis confirmed that LC3-II levels increased, while p62/SQSTM1 levels decreased, in a time-dependent manner in both glioma cells upon exposure to RSL3 or erastin (Fig. 3a and Supplementary Fig. 3a). Moreover, cell death caused by RSL3 or erastin was dampened by the autophagy inhibitors bafilomycin A1 (BafA1) and hydroxychloroquine (HCQ), indicating that RSL3 or erastin might trigger autophagy-associated cell death in glioma cells (Fig. 3b and Supplementary Fig. 3b). BafA1 is widely used in vitro as an autophagic flux inhibitor, thereby interfere with lysosome-mediated degradation of autophagic proteins such as LC3-II and p62/SQSTM1. However, the accumulation of LC3-II may signify either an increase in autophagosome formation and/or a block of autophagic flux. To distinguish between the two possibilities, we performed LC3-II and p62/SQSTM1 turnover assays using BafA1 to block autophagosome fusion with lysosomes. As expected, BafA1 can further enhance RSL3-induced LC3-II accumulation and inhibit p62/SQSTM1 degradation in glioma cells (Fig. 3c and Supplementary Fig. 3c), indicating that autophagic flux is increased during ferroptosis. Additionally, we used the ptfLC3 (tandem reporter mRFP-GFP-LC3) construct to study the role of RSL3 in autolysosome formation. This dual-tagged LC3 construct can monitor the maturation

process of autophagosomes and autolysosomes with different fluorescent signals[6]. As shown in Fig. 3d, U87MG cells treated with RSL3 exhibited both yellow (representing autophagosomes) and red (representing autolysosomes) puncta formation, both of which were not detected in the cells of the control group. In contrast, BafA1 (which inhibits acidification inside the lysosome and thus impairs autolysosomal maturation) augmented only the yellow puncta (Fig. 3d), indicating a close association between autophagy and ferroptosis. To establish the functional link between autophagy and ferroptosis, we inhibited autophagy by pharmacological and genetic approaches. We found that the pharmacological inhibition of autophagy by BafA1 or HCQ significantly blocked RSL3- and erastin-induced GPX4 degradation (Fig. 3c and Supplementary Fig. 3c), labile iron accumulation (Fig. 3e and Supplementary Fig. 3d, e), and lipid peroxidation (Fig. 3f and Supplementary Fig. 3f). Moreover, we observed that shRNA (Supplementary Fig. 3g) or siRNA (Supplementary Fig. 3h)-mediated knockdown of two vital autophagy-related genes (Atg-5 and Atg-7)[46] significantly attenuated RSL3-induced LC3-II conversion (Fig. 3g), p62/SQSTM1 (Fig. 3g), GPX4 degradation (Fig. 3g), labile iron accumulation (Fig. 3h), lipid peroxidation (Fig. 3i), and cell death (Fig. 3j), collateral that increased autophagy contributes to the ferroptotic cell death.

Our previous studies have shown that Par-4 plays a crucial role in modulating autophagic cell death in human glioblastoma cells[29,30]. To ascertain whether Par-4 regulated ferroptosis via an autophagy-dependent mechanism, we measured the LC3-II conversion and p62/SQSTM1 degradation in Par-4 knockdown cells. Genetic deletion of Par-4 using shRNA, CRISPR or siRNA inhibited LC3-II conversion and p62/SQSTM1 degradation in glioma cells induced by RSL3 and ML210, and erastin (Fig. 3k–m and Supplementary Fig. 3i, j). Consistent results were also observed in Par-4$^{-/-}$ MEFs (Supplementary Fig. 3k), suggesting that Par-4-meditated autophagy plays a critical role in ferroptosis. Next, we examined whether Par-4 overexpression potentiates RSL3-induced autophagy. RSL3 treatment of Par-4-overexpressing glioma cells showed a marked increase in LC3-II conversion and p62/SQSTM1 degradation (Fig. 3n, o). Similar results were obtained in DOX-induced Par-4-U87MG cells (Supplementary Fig. 3l). Surprisingly, Par-4 overexpression alone was sufficient to induce autophagosome formation, as evidenced by increased LC3-II expression and p62/SQSTM1 degradation (Fig. 3n, o and Supplementary Fig. 3k, l), indicating that Par-4 itself might play an important role in facilitating autophagy-mediated ferroptosis.

Additionally, we tested whether endogenous Par-4 increases autophagic flux following BafA1 administration. The levels of LC3-II and

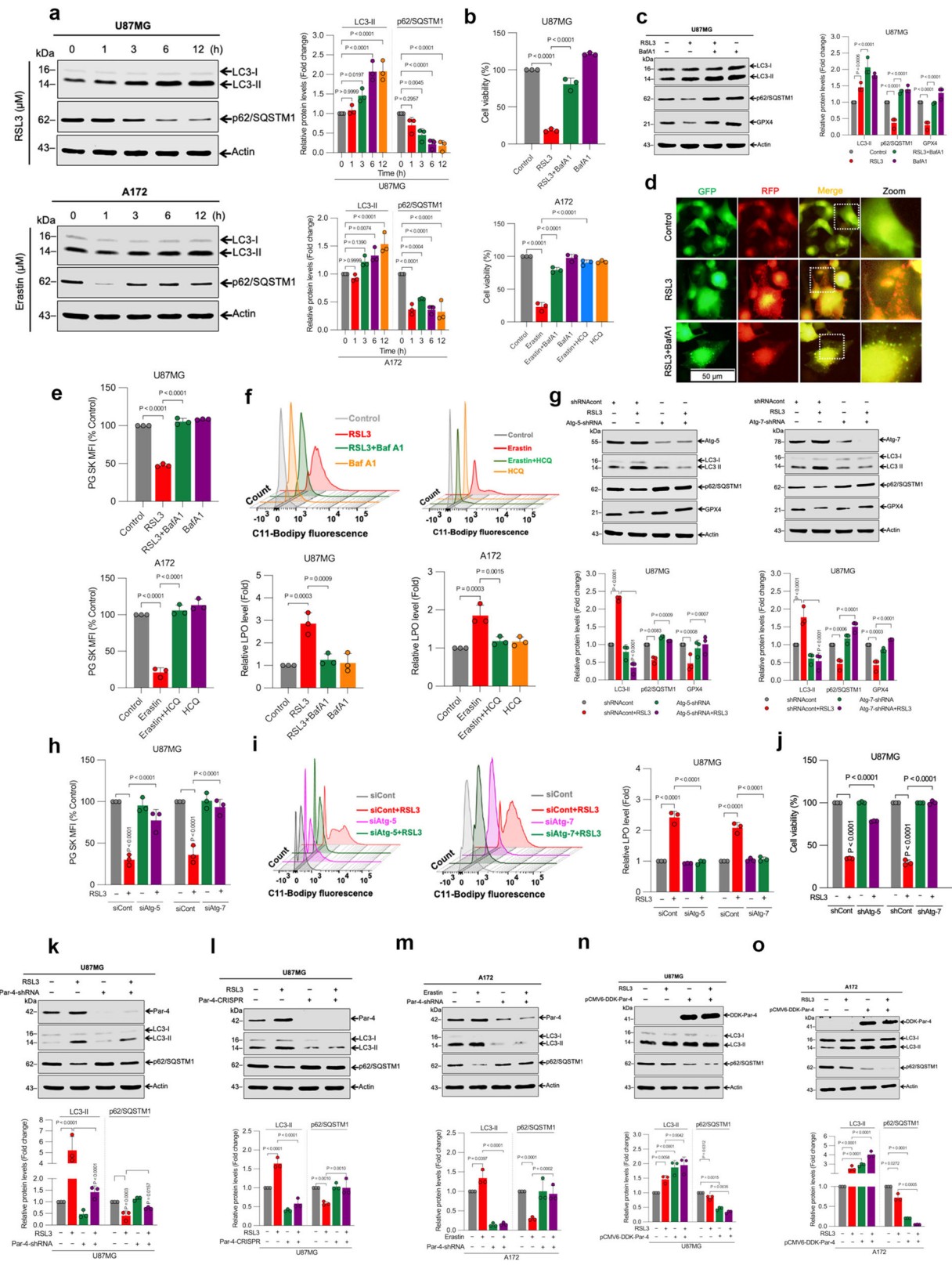

p62/SQSTM1 were increased in BafA1-treated empty vector cells. However, the addition of RSL3 in empty vector cells further potentiated the autophagic flux than in cells treated with BafA1 alone (Supplementary Fig. 3m). Interestingly, overexpression of Par-4 in the presence of BafA1 further enhanced RSL3-induced autophagic flux. It is important to note that BafA1

prevented the reduction of GPX4 protein levels induced by RSL3 in both empty vectors and endogenous Par-4 vector cells, under the same treatment conditions (Supplementary Fig. 3m). Collectively, these findings strongly suggest that Par-4 plays a significant role in the autophagic degradation of GPX4, which ultimately leads to ferroptosis.

**Fig. 3 | Par-4 regulates ferroptosis through autophagy. a** Indicated cells were treated with RSL3 (2 μM) and erastin (10 μM) for the indicated time period. Following the treatment, Western blot analysis of LC3 and p62/SQSTM1 were detected. The relative density of protein bands were quantified and normalized to the actin of each group, and fold changes were presented in histogram from three independent experiments. Indicated cells were treated with RSL3 (2 μM) and erastin (10 μM) for 24 h in the presence or absence of BafA1 (250 nM) or HCQ (5 μM). Following the treatment, **b** Cell viability by MTT assay. Data shown are mean ± SD; $n = 3$ samples, and **c** Western blot analysis of indicated proteins were measured. The relative density of protein bands were quantified and normalized to the actin of each group, and fold changes were presented in histograms from three independent experiments. **d** U87MG cells expressing ptfLC3 were treated with RSL3 (2 μM) for 6 h. Representative fluorescent images were captured at x20 magnification using fluorescent microscopy. BafA1 (250 nM) was used as a negative control for autolysosome formation. Yellow puncta represent the number of autophagosomes, and red puncta in merged images represent autolysosomes. Indicated cells were treated with RSL3 (2 μM) for 3 h and erastin (10 μM) for 24 h in the presence or absence of BafA1 (250 nM) or HCQ (5 μM). Following the treatment, **e** Intracellular labile iron was determined using flow cytometry. The bar graph showing labile iron level was expressed as a percentage of the control, and **f** Lipid peroxidation was detected by flow cytometry. Bar graph showing the relative levels of lipid peroxidation. Data shown are mean ± SD; $n = 3$ samples. **g** U87MG cells were stably transfected with Atg-5-shRNA or Atg-7-shRNA, followed by exposure to RSL3 (2 μM) for 3 h. Following the treatment, a Western blot analysis of indicated proteins were performed. The relative density of protein bands were quantified and normalized to the actin of each group, and fold changes were presented in histograms from three independent experiments. U87MG cells were transiently transfected with siAtg-5 and siAtg-7, followed by exposure to RSL3 (2 μM) for 3 h. Following the treatment, **h** intracellular labile iron was determined using flow cytometry. The bar graph showing labile iron level was expressed as a percentage of the control, and **i** Lipid peroxidation was detected by flow cytometry. Bar graph showing the relative levels of lipid peroxidation. Data shown are mean ± SD; $n = 3$ samples. **j** U87MG cells were stably transfected with Atg-5-shRNA or Atg-7-shRNA. After transfection, cells were treated with RSL3 (2 μM; Atg-5-shRNA and 0.5 μM; Atg-7-shRNA) for 24 h. Following the treatment, cell viability by MTT assay was assessed. Data shown are mean ± SD; $n = 3$ samples. U87MG cells were stably transfected with **k** Par-4-shRNA and **l** Par-4-CRISPR. After transfection, cells were treated with RSL3 (2 μM) for 3 h, and a Western blot analysis of indicated proteins was carried out. The relative density of protein bands were quantified and normalized to the actin of each group, and fold changes were presented in histograms from three independent experiments. **m** A172 cells were stably transfected with Par-4-shRNA. After transfection, cells were treated with erastin (10 μM) for 24 h, and a Western blot analysis of indicated proteins was carried out. The relative density of protein bands were quantified and normalized to the actin of each group, and fold changes were presented in histograms from three independent experiments. **n** U87MG and **o** A172 cells were transiently transfected with pCMV6-DDK-Par-4. After transfection, cells were treated with RSL3 (2 μM) for 3 h. Following the treatment, a Western blot analysis of indicated proteins were carried out. The relative density of protein bands were quantified and normalized to the actin of each group, and fold changes were presented in histograms from three independent experiments. Data shown are mean ± SD; $n = 3$ samples. Statistical significance (*P* values) was analyzed by one-way ANOVA using the Bonferroni post-hoc test.

## Par-4 regulates ferroptosis by inducing ferritinophagy

Ferritinophagy, a selective form of autophagy, contributes to ferroptosis induction through the autophagic degradation of ferritin, a major intracellular iron storage protein. Mechanistically, NCOA4 is a ferritinophagy receptor, which interacts with an arginine residue in the C terminal domain of ferritin heavy chain 1 (FTH1), accounting for selective sequestration and degradation of ferritin. Induction of ferritinophagy triggers labile iron accumulation, ROS generation, lipid peroxidation, and cell death[44,47,48]. Thus, we examined whether ferritinophagy is the primary mechanism involved during RSL3- and erastin-induced ferroptosis in glioma cells. First, we determined the effects of treatment with RSL3 or erastin on expression levels of FTH1 and NCOA4. Consistent with the previous reports[7,47,49], RSL3, and erastin induced the time-dependent degradation of FTH1 and NCOA4 in glioma cells (Fig. 4a and Supplementary Fig. 4a). Interestingly, pre-treatment with DFO significantly inhibited RSL3-induced FTH1 degradation (Fig. 4b). In addition to FTH1 degradation, RSL3 significantly elevated intracellular labile iron levels. This increase was suppressed by co-treatment with DFO (Fig. 4c and Supplementary Fig. 4b). Consistent findings were also corroborated with FerroOrange dye. RSL3 treatment increased intracellular labile iron levels, as seen by the red-orange fluorescence accumulated in the cells. Pre-treatment with the iron chelators ciclopirox (CPX) and DFO blocked the red-orange fluorescence, as expected (Fig. 4d). The FerroOrange median fluorescence intensity (MFI) was further quantified using flow cytometry (Fig. 4e). Together, these results suggest that ferritinophagy-mediated alterations in ferritin and NCOA4 level increase the labile iron pool, leading to ferroptosis in glioma cells. Next, we sought to clarify whether inhibition of autophagy might suppress the FTH1 and NCOA4 degradation. Remarkably, both pharmacological (BafA1) (Fig. 4f) and genetic (shAtg-5 and shAtg-7) (Fig. 4g) inhibition of autophagy significantly diminished RSL3-induced FTH1 and NCOA4 degradation, indicating that autophagy mediates ferritin degradation in glioma cells during ferroptosis.

Our aforementioned data prompted us to assess whether the upregulation of Par-4 was directly involved in the activation of ferritinophagy during ferroptosis. Western blot analysis revealed that the genetic inhibition of Par-4 by shRNA, CRISPR, or siRNA, significantly reversed the RSL3, ML210, or erastin-induced FTH1 and NCOA4 degradation (Fig. 4h, i, Supplementary Figs. 3j, 4c, d). Furthermore, consistent with glioma cells, double knockout Par-4[−/−] MEFs also significantly inhibited RSL3 or ML210-induced degradation of FTH1 and NCOA4 (Fig. 4j and Supplementary Fig. 4e). On the other hand, overexpression of Par-4 by DDK-Par-4 (Fig. 4k, l) and DOX-inducible Par-4 (Supplementary Fig. 4f) remarkably enhanced RSL3-induced FTH1 and NCOA4 degradation and promoted ferroptosis. Collectively, our results provide direct evidence linking Par-4 to NCOA4-mediated degradation of ferritin during ferroptosis.

## Par-4 intensifies ROS production during ferroptosis

As mentioned above, ferritin degradation increases cellular free iron levels and triggers the Fenton reaction to promote ROS accumulation and cell death[50]. Therefore, we measured intracellular ROS levels in RSL3-treated cells using 2′,7′-dichlorodihydrofluorescein diacetate (DCFH-DA), an oxidation-sensitive fluorescent probe. Our results show that RSL3 markedly increased intracellular ROS production, and this effect was inhibited by lipid peroxidation scavengers, Lip-1, Fer-1, and iron chelator DFO (Fig. 5a), suggesting that ROS accumulation is critical during ferroptosis. Next, we determined the role of Par-4 activation in RSL3-induced ROS accumulation. As shown in Fig. 5b and Supplementary Fig. 5, RSL3-induced ROS bursts in glioma cells were significantly inhibited by siRNA-mediated Par-4 downregulation. On the other hand, overexpression of Par-4 amplified RSL3-induced ROS production (Fig. 5c). Similar results were observed in Par-4[+/+] and Par-4[−/−] MEFs (Fig. 5d), suggesting that Par-4 is essential for RSL3-induced ROS production, an event critical for the induction of ferroptosis.

The data presented above indicate that Par-4 controls ROS generation upon RSL3-induced ferroptosis. Redox regulation of Par-4 has been well-documented in our previous studies[29,51]. Therefore, we checked if there is a positive feedback regulation between Par-4 activation and ROS generation during RSL3-induced ferroptosis. Pre-treatment with ROS scavengers N-acetylcysteine (NAC) and Trolox (6-hydroxy-2,5,7,8-tetramethylchromane-2-carboxylic acid) attenuated RSL3-induced Par-4 induction (Fig. 5e), LC3-II conversion, and p62/SQSTM1, FTH1, and NCOA4 degradation in glioma cells (Fig. 5f–h). Together, these results indicate that Par-4 and ROS molecules regulate each other in a positive feedback manner during RSL3-induced autophagy-mediated ferroptotic cell death in GBM cells.

## Par-4 regulates ferroptosis in tumor xenograft in vivo

Finally, we examined whether the downregulation of Par-4 modulates the antitumor activity of RSL3 in a xenograft mouse model. Control shRNA and

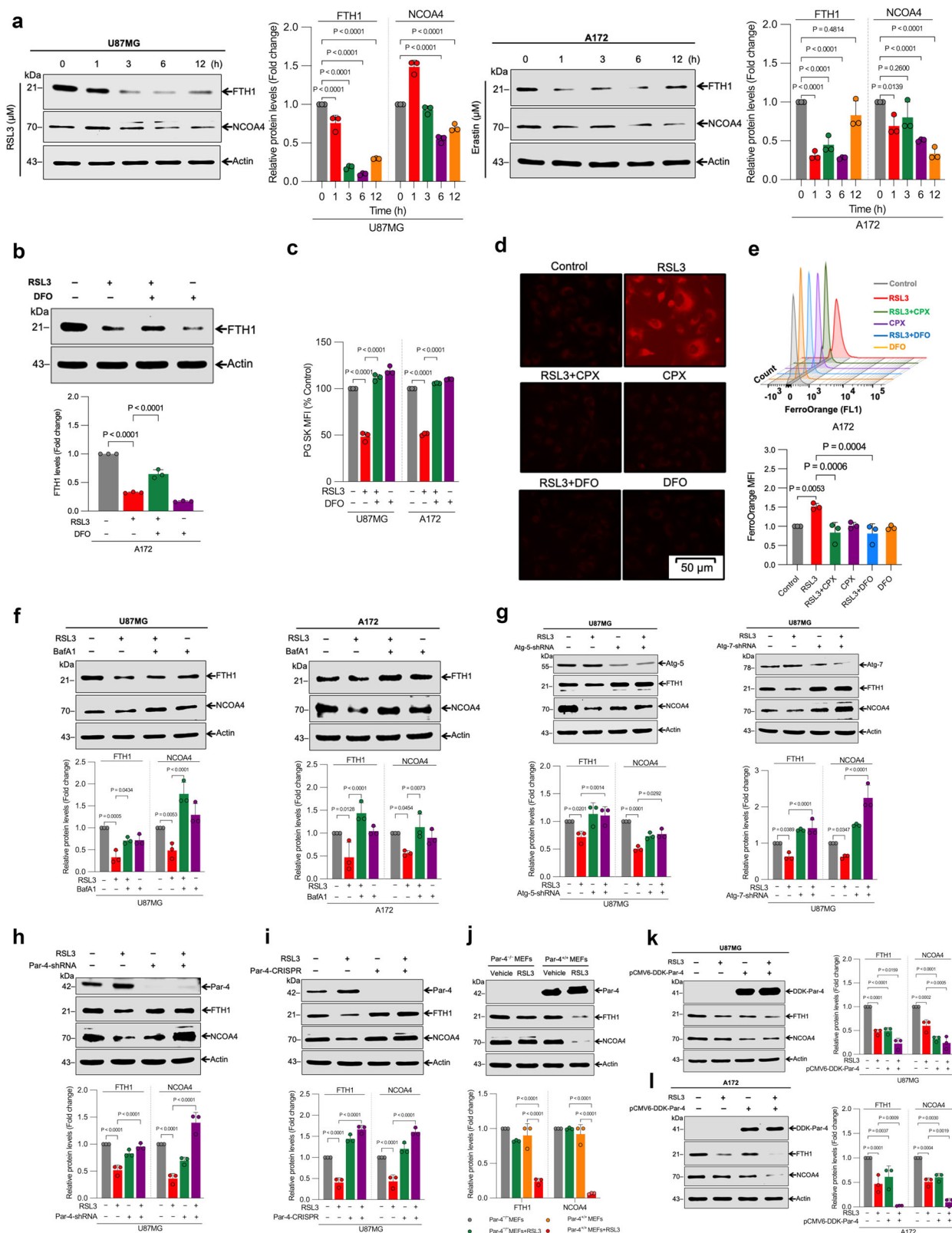

shRNA-mediated Par-4-knockdown U87MG cells were implanted subcutaneously into the right flank of immunodeficient nude mice. A schematic of tumor inoculation and systemic drug injection (i.p) is shown in Fig. 6a, b. Once the tumor volume reached ~25 mm³, the mice were randomly allocated into groups ($n = 4$) and treated with vehicle (0.02% DMSO) or RSL3 (4.4 mg/kg)[36,52–56]. None of the treatments adversely affected the body weight of the mice (Fig. 6c). RSL3 treatment strongly suppressed the growth of control shRNA tumors, but the ferroptosis activator had a negligible effect on the growth of Par-4-knockdown tumors (Fig. 6d–g). Likewise, tumor mass analysis after 21 days of treatment revealed a significant effect of RSL3 treatment on control shRNA, but not Par-4-knockdown, U87MG xenografts (Fig. 6e, f). Western blot analysis confirmed the efficiency of Par-4

**Fig. 4 | Par-4 is required for the NCOA4-mediated degradation of ferritin during ferroptosis. a** Indicated cells were treated with RSL3 (2 μM) and erastin (10 μM) for the indicated time period, and then Western blot analysis of FTH1 and NCOA4 were performed. The relative density of protein bands were quantified and normalized to the actin of each group, and fold changes were presented in histograms from three independent experiments. **b** A172 cells were pre-treated with DFO (50 μM) for 1 h, followed by RSL3 (2 μM) for a further 3 h. Following the treatments, a Western analysis of FTH1 was carried out. The relative density of protein bands was quantified and normalized to the actin of each group, and fold changes were presented in histograms from three independent experiments. **c** Indicated cells were pre-treated with DFO (100 μM) for 1 h, followed by RSL3 (2 μM) treatment for a further 3 h. After treatment, intracellular labile iron was determined using flow cytometry. The bar graph showing labile iron level was expressed as a percentage of the control. Data shown are mean ± SD; n = 3 samples. **d** A172 cells were treated with RSL3 (2 μM) for 3 h in the presence or absence of CPX (10 μM) or DFO (100 μM). After the treatment, the cells were subjected to FerroOrange staining to evaluate intracellular LIP. Red fluorescence signals were captured and visualized through a fluorescent microscope using constant fluorescence parameters explained in the methods section: scale bar, 50 μm. **e** Subsequently, median fluorescence intensity (MFI) was quantified by flow cytometry analysis. The bar graph shows relative levels of LIP by FerroOrange staining in the indicated cells. Data shown are mean ± SD;

n = 3 samples. Indicated cells were treated with RSL3 (2 μM) in the presence or absence of **f** BafA1 (250 nM) and **g** Atg-5-shRNA or Atg-7-shRNA. Following the treatment, a Western blot analysis of indicated proteins were analyzed. The relative density of protein bands were quantified and normalized to the actin of each group, and fold changes were presented in histograms from three independent experiments. U87MG cells were stably transfected with **h** Par-4-shRNA and **i** Par-4-CRISPR. After transfection, cells were treated with RSL3 (2 μM) for 3 h, and a Western blot analysis of indicated proteins was carried out. The relative density of protein bands were quantified and normalized to the actin of each group, and fold changes were presented in histogram from three independent experiments. Par-4[+/+] and Par-4[−/−] MEFs were treated with RSL3 (2 μM) for 3 h. Following the treatment, **j** Western blot analysis of indicated proteins were carried out. The relative density of protein bands were quantified and normalized to the actin of each group, and fold changes were presented in histograms from three independent experiments. **k** U87MG and **l** A172 cells were transiently transfected with pCMV6-DDK-Par-4. After transfection, cells were treated with RSL3 (2 μM) for 3 h. Following the treatment, a Western blot analysis of indicated proteins were carried out. The relative density of protein bands were quantified and normalized to the actin of each group, and fold changes were presented in histogram from three independent experiments. Data shown are mean ± SD; n = 3 samples. Statistical significance (P values) was analyzed by one-way ANOVA using the Bonferroni post-hoc test.

knockdown in U87MG cells isolated from RSL3-treated xenografts (Supplementary Fig. 6a). Notably, degradation of FTH1 was observed in RSL3-treated shRNA control, but not Par-4 knockdown, U87MG xenografts (Fig. 6h), suggesting that Par-4 is required for ferritinophagy-mediated ferroptosis in vivo. We then performed the Histological examination of tumor tissue sections using hematoxylin and eosin (H&E) staining, confirmed that the RSL3 treatment induced significant cell death in shRNA control U87MG tumors, but the ferroptosis activator did not significantly adversely affect Par-4 knockdown U87MG tumors (Fig. 6i). Importantly, H&E staining of vital organ (heart, lungs, liver, spleen, and kidney) sections showed no abnormalities and lesions (Supplementary Fig. 6b). These findings are consistent with the in vitro results indicating that Par-4 contributes to the core molecular machinery and signaling pathway of ferroptosis-mediated tumor suppression.

## Discussion

Although apoptosis, senescence, and metastasis-inducing tumoricidal functions of Par-4 have been significantly studied, the role and mechanism of action of Par-4 in the induction of ferroptosis have never been explored[18,30–32]. In this study, we reveal that Par-4 plays a significant role in regulating ferroptosis. Mechanistically, Par-4-mediated redox amplification contributes to the autophagic degradation of FTH1, ultimately leading to labile iron overload, and devastating lipid peroxidation, which are the major driving events of ferroptosis (Fig. 7).

In this study, through RNA-Seq screening, Par-4 is identified to be activated during the execution of ferroptosis. Moreover, we have revealed that classical ferroptosis stimuli, such as RSL3, ML210, or erastin, trigger selective activation of Par-4, leading to significant lipid peroxidation in GBM cells. Par-4-mediated lipid peroxidation parallels the sensitivity of glioma cells to FINs based on levels of ferroptosis markers, such as GPX4 degradation, labile iron overload, ROS generation, and cell death. Par-4 activation of a similar magnitude has previously been reported in ceramide and arsenic trioxide-induced autophagic cell death in glioma cells[30]. Simultaneously, Par-4 activation has also been reported in curcumin-induced autophagic cell death of malignant glioma cells[29]. Our data demonstrate that the knockdown of Par-4 using siRNA, shRNA, and CRISPR-Cas9 technology significantly attenuated RSL3 or erastin-induced GPX4 degradation, labile iron overload, ROS generation, lipid peroxidation, and ferroptosis. Interestingly, our results indicate that endogenous Par-4 is sufficient to induce ferroptosis, as evidenced by basal levels of GPX4 being further decreased with or without RSL3 treatment. Our current findings are consistent with other reports that endogenous Par-4 is crucial for inducing apoptosis[16,27].

Initially, ferroptosis was defined as a unique independent cell death process[57]. Later, it was found that classical FINs require autophagy machinery to execute ferroptosis[5,7,8,11]. Indeed, our results strongly support the latter theory, demonstrating that autophagy is crucial for RSL3- and erastin-induced GBM cell death. Previously, Par-4 has been shown to play a vital role in executing autophagy initiated by various cytotoxic agents and other anticancer drugs[29,30]. Additionally, it has been evident that Par-4 induces concomitant apoptosis and autophagic cell death in hypopharyngeal carcinoma cells[58]. Our data prove that Par-4 plays a significant role in the autophagic degradation of GPX4, which ultimately leads to ferroptosis. To date, several selective forms of autophagy, including ferritinophagy, clockophagy, lipophagy, mitophagy, and chaperone-mediated autophagy, have been identified as participants in ferroptotic cell death by degrading anti-ferroptosis proteins or organelles[5,7–9]. In particular, ferritinophagy refers to the selective autophagic degradation of ferritin. Furthermore, ferritinophagy is crucial in maintaining iron homeostasis, while excessive ferritinophagy contributes to iron overload and ferroptosis[59,60]. Previously, Zhang et al. demonstrated that erastin- and sorafenib-induced-ELAV-like RNA binding protein 1 (ELAVL1) upregulation promotes autophagy, autophagosome accumulation, ferritinophagy, and ferroptosis in hepatic stellate cells[44]. More recently, Chen et al. also have reported that HPCAL1 (hippocalcin-like 1) is a membrane-enriched protein that mediates ferroptosis through autophagy[61]. In line with these findings, we demonstrate a distinct mechanism through which Par-4 regulates ferritinophagy. Genetic deletion of Par-4 significantly protected cells from FINs-induced ferritinophagy and ferroptosis. Meanwhile, overexpression of Par-4 profoundly sensitized these cells to FINs-induced ferritinophagy and ferroptosis. These findings suggest that Par-4 plays a pivotal role in the induction of autophagy-mediated ferroptosis. It is worth noting that Par-4 promotes not only classical autophagy but also ferroptosis-related selective autophagy in GBM cells. As mentioned earlier, ferritinophagy, a specialized form of autophagy, targets intracellular ferritin for degradation primarily within lysosomes, releasing free iron and inducing ferroptosis[59,60]. The precise interplay between Par-4 and ferroptosis, particularly their involvement in ferritinophagy, remains an active area of research. Our results indicate a connection between Par-4 and ferritin degradation via NCOA4 during ferroptosis. However, additional investigations are required to elucidate how Par-4 regulates NCOA4/FTH1 degradation within lysosomes, supplementing our immunoblot confirmation.

Several reports suggest that RSL3 or its structurally modified analog, administered intraperitoneally (i.p), was used in vivo for xenograft models[52–56]. We further confirmed the role of Par-4 in ferroptosis-mediated tumor suppression by utilizing Par-4 knockout U87MG xenograft models.

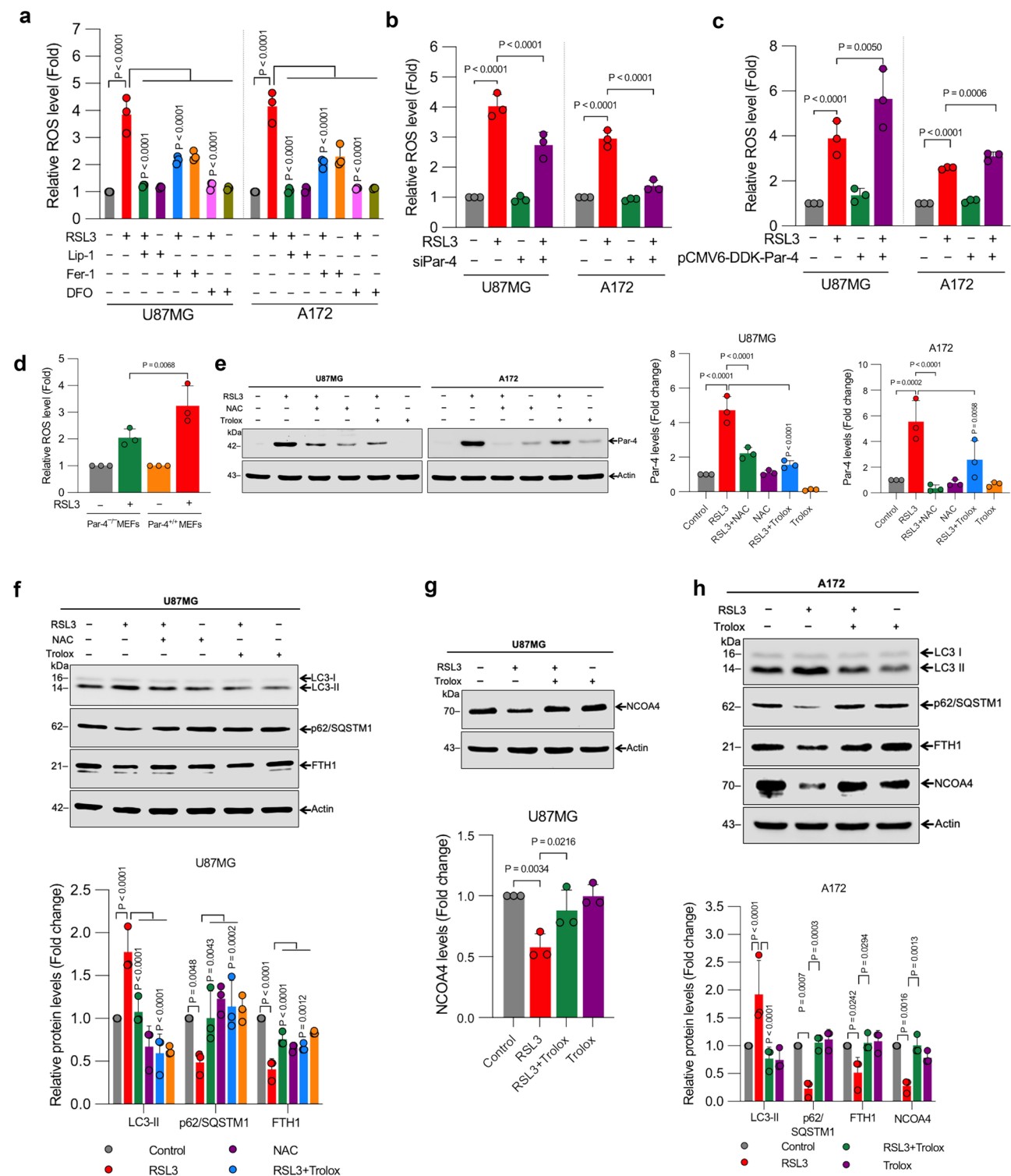

**Fig. 5 | Par-4-dependent ROS accumulation is critical for the ferritinophagy–ferroptosis axis. a** Indicated cells were treated with RSL3 (2 μM) for 24 h in the presence or absence of Lip-1 (1 μM), Fer-1 (5 μM), and DFO (100 μM). ROS generation was quantified. Data shown are mean ± SD; ($n = 3$). **b** Par-4 depleted or **c** Par-4 overexpressed cells were treated with RSL3 (2 μM) for 3 h, and then ROS generation was quantified. Data shown are mean ± SD; $n = 3$ samples. **d** Par-4$^{+/+}$ and Par-4$^{-/-}$ MEFs were treated with RSL3 (2 μM) for 3 h, and ROS generation was measured. Data shown are mean ± SD; $n = 3$ samples. **e** Indicated cells pre-treated with NAC (2.5 mM) and Trolox (1 mM) were treated with RSL3 (2 μM). Following the treatment, cells were analyzed for Western blot analysis of Par-4. The relative

density of protein bands were quantified and normalized to the actin of each group, and fold changes were presented in histograms from three independent experiments. **f–h** Indicated cells pre-treated with NAC (2.5 mM) or Trolox (1 mM) were treated with RSL3 (2 μM). Following the treatment, cells were analyzed for Western blot analysis of indicated proteins. The relative density of protein bands were quantified and normalized to the actin of each group, and fold changes were presented in histograms from three independent experiments. Data shown are mean ± SD; $n = 3$ samples. Statistical significance (*P* values) was analyzed by one-way ANOVA using the Bonferroni post-hoc test.

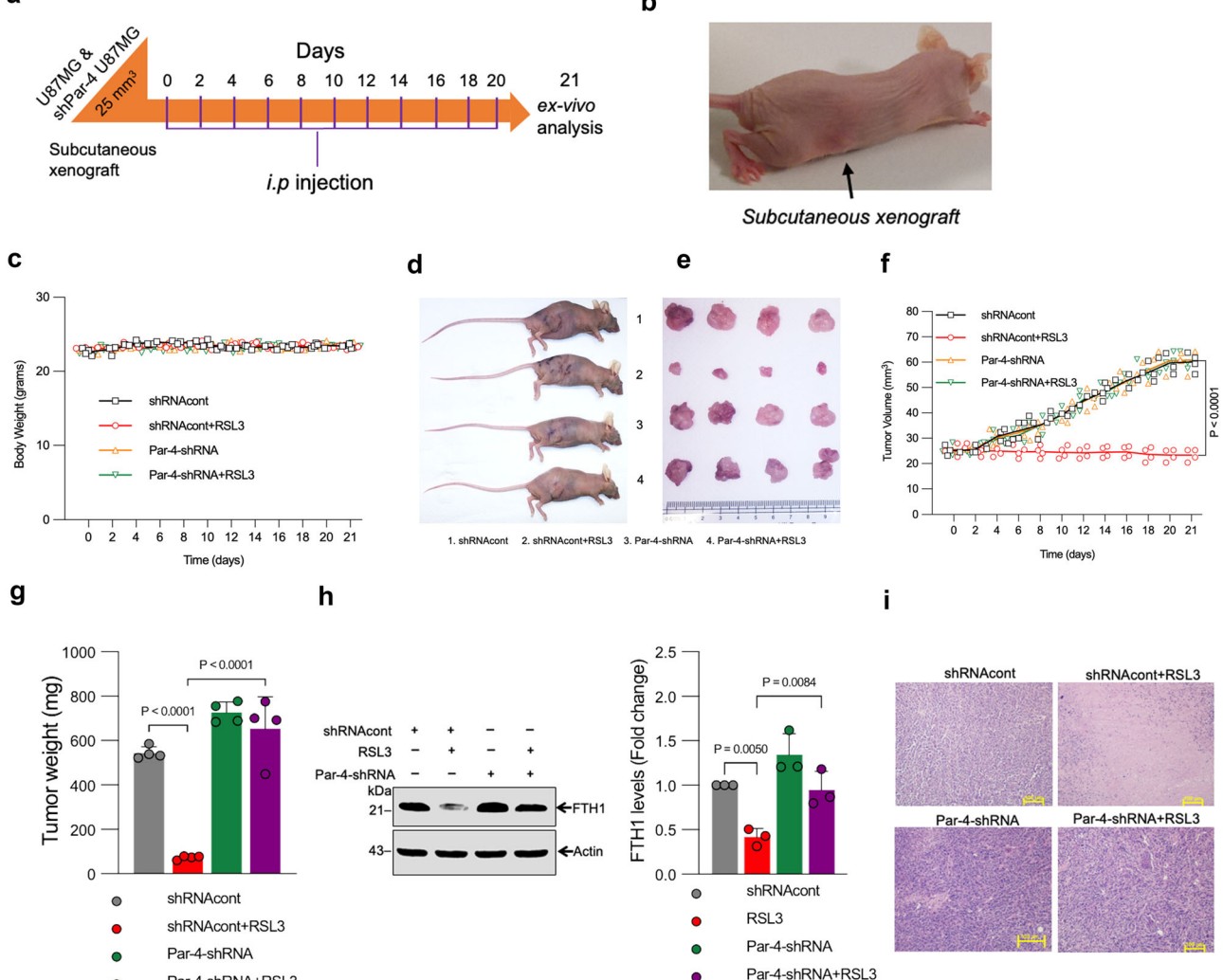

**Fig. 6 | Par-4 regulates ferroptosis in tumor xenografts in vivo.** The methods section describes the experimental plan and detailed tumor implantation and treatment groups. **a** Design of the tumor reduction studies. BALB/c nude mice were subcutaneously implanted with the indicated Par-4-knockdown (shPar-4 U87MG) or control U87MG (shRNAcont) cells for 7 days and then treated with RSL3 (4.4 mg/kg, i.p., three times per week for 21 days). **b** A representative mouse was bearing subcutaneous xenograft (black color arrow indicated). **c** Body weight changes of the tumor-bearing mice in the different treatment groups were monitored for the duration of the experiment (mean ± SD; $n = 4$). **d** Photographs of mice with tumor xenografts. Tumor volume growth curves for the shRNA control and Par-4 stable knockdown U87MG xenografts over 21 days of treatment. After 21 days of treatment, four mice per treatment group were sacrificed, and the tumor tissues were isolated and imaged **d**, **e**; tumor volume was calculated using the formula defined in methods **f** and **g** subsequently weighed to determine the tumor mass. Data shown are mean ± SD; ($n = 4$ mice/group). **h** Western blot analysis of FTH1 expression in isolated tumors from the U87MG xenograft model were determined. The relative density of protein bands were quantified and normalized to the actin of each group, and fold changes were presented in histograms from three independent experiments. **i** Hematoxylin and eosin (H&E)-stained xenograft sections from the different treatment groups following 21 days. The images shown represent tissue sections from four mice per treatment group; Scale bar, 100 μm. Statistical analysis was performed using Two-way ANOVA followed by Tukey's post hoc test between control and treatment groups.

These findings imply that Par-4 could hold therapeutic potential for treating human glioblastoma. With the increasing recognition of the fundamental role of ferritinophagy in ferroptosis, and the possible contribution of ferroptosis to many pathological conditions, such as neurodegeneration and ischemia-reperfusion injury, strategies aiming at the inhibition of ferritinophagy are emerging as attractive cytoprotective strategies[2]. Therefore, identifying Par-4 as an essential ferroptosis regulator is important in this context. It is necessary and exciting to explore this question further. Another significant finding from the current study is that ROS accumulation is required for autophagy-dependent ferroptosis. ROS production is a well-known direct trigger of ferroptosis[57,62,63]. Our results reveal that inhibition of ROS could significantly attenuate the effects of RSL3 on both autophagy and ferritinophagy, indicating that ROS accumulation is critical for RSL3-induced autophagic turnover of ferritin and ferroptosis in GBM cells.

Previously, we reported that curcumin-induced Par-4 expression was mediated by ROS. In that study, we found that ROS scavengers, such as GSH and NAC, significantly abolished the induction of Par-4 by curcumin[29]. Further, we have shown that extracellular supplementation of $H_2O_2$ resulted in significant induction of Par-4 expression in both U87MG and U118MG glioma cells. In addition, we have shown that ROS is involved in the cleavage of Par-4. In both PC3 and DU145 prostate cancer cells, pre-treatment with Sod Pyr, a well-established $H_2O_2$ scavenger, completely inhibited Par-4-mediated cell death[51]. Interestingly, in the present study, ROS scavengers, such as NAC and Trolox, abolished the effect of RSL3 on Par-4 levels, suggesting that ROS induction is a mechanism contributing to RSL3-induced Par-4 activation. Together our results indicate that Par-4 and ROS molecules are positively regulated by one another during RSL3-induced ferritinophagy activation to accelerate ferroptosis.

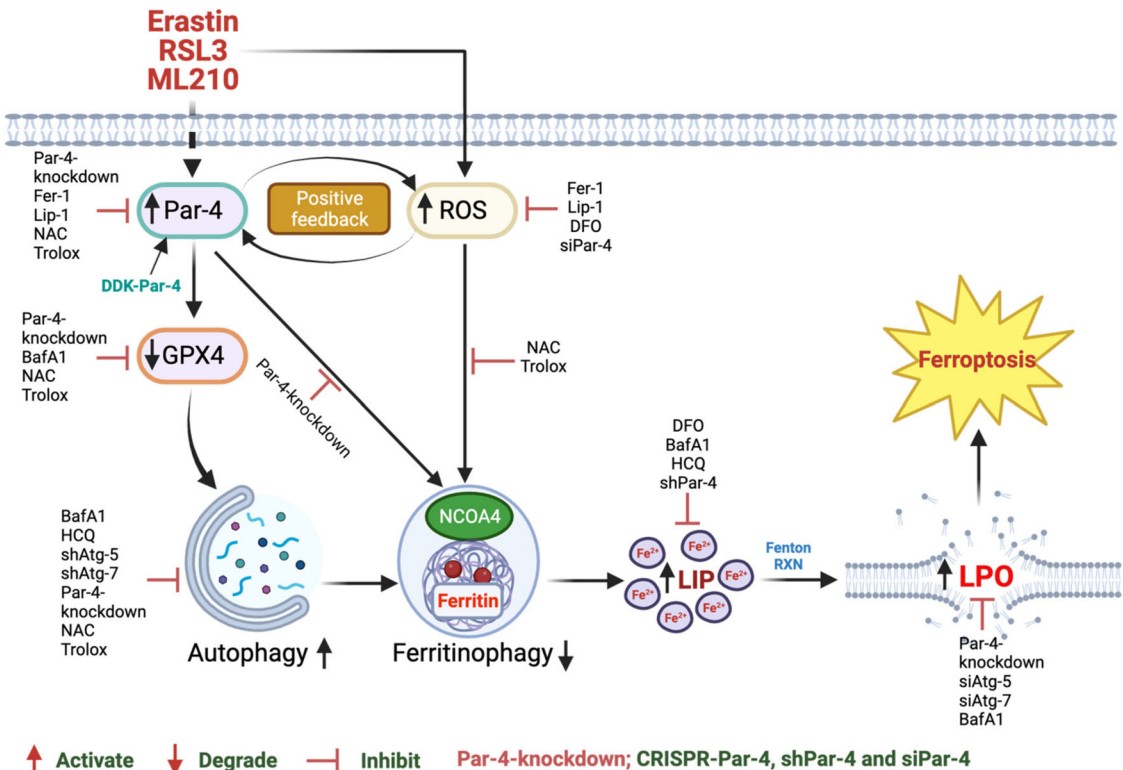

**Fig. 7 | Schematic representation of Par-4-dependent ROS accumulation plays a critical role in RSL3, ML210-and erastin-induced autophagy-dependent ferroptosis in human glioblastoma cells.** Created with BioRender.com (Agreement number: NY26BEEWLM).

Collectively, our results delineate a distinctive mechanistic pathway involving Par-4-mediated ferritinophagy, reactive oxygen species (ROS) responses, and subsequent induction of ferroptotic cell death. The elucidation of this pathway holds promise for therapeutic exploitation, particularly in the context of pathological conditions that link to ferroptosis.

## Methods
### Reagents and antibodies
Erastin (#E7781), (1 S, 3 R)-RSL3 (#SML2234), MTT (3-[4,5-dimethyl-thiazol-2-yl]-2,5-diphenyl tetrazolium bromide; #M5655), sodium chloride (#S3014), NAC (#A9165), Trolox (#238813), Bisbenzimide (Hoechst 33342; #B2883), crystal violet (#32675), Lip-1 (#SML1414), HCQ (#H0915), dimethyl sulfoxide (DMSO; #5879, #D4259), DCFH-DA (#D6883), phenylmethylsulfonyl fluoride (PMSF; #P7626), and dithiothreitol (DTT; #D9779) were purchased from Sigma Chemical Co. (St. Louis, MO, USA). ML210 (#S0788) was purchased from Selleck Chemicals LLC, Houston, USA. Fer-1 (#SC-498126), DFO (#SC-203331), and BafA1 (#SC-201550) were purchased from Santa Cruz Biotechnology Inc. (Santa Cruz, CA, USA). Image-iT™ lipid peroxidation kit (#C10445) and PG SK (#P14313) were purchased from Thermo Fisher Scientific, Grand Island, NY. FerroOrange (#F374-12) was purchased from Dojindo EU GmbH, Germany. Necrosulfonamide (NSA; #490073) and GSK′872 (#530389) were purchased from Calbiochem, San Diego, CA, USA. GSK′963 (#GLXC-08006) was purchased from Glixx Laboratories, Hopkinton, MA, USA. N-benzoyloxycarbonyl-Val-Ala-Asp fluoro-methyl ketone (z-VAD-FMK; #ALX260-020-M005) was purchased from Enzo Life Sciences, Switzerland. MegaTran 1.0 (#TT200002) was purchased from Origene Technologies, Rockville, MD, USA. HiPerFect transfection reagent (#301705) was purchased from Qiagen, Valencia, CA, USA. Puromycin (#ANT-PR-1) was purchased from InvivoGen, San Diego, CA, USA. Dulbecco's modified essential medium (DMEM; #31885), phosphate-buffered saline (PBS; #14190-094), trypsin-EDTA (#25300-054), Penicillin Streptomycin (#15140-122), and fetal bovine serum (FBS; #10270) were purchased from Gibco BRL (Grand Island, NY, USA).

### Cell culture conditions and drug treatment
Human glioblastoma cell line U87MG was obtained from ATCC (Rockville, MD, USA), and the A172 cell line was obtained from the European Collection of Authenticated Cell Cultures (ECACC) (Porton Down, Salisbury, UK). Par-4$^{+/+}$ and Par-4$^{-/-}$, primary mouse embryonic fibroblasts (MEFs) cells were kindly provided by Prof. Vivek Rangnekar, Department of Radiation Medicine and Markey Cancer Center, University of Kentucky, Lexington, KY, USA. The protocol for Par-4$^{+/+}$ and Par-4$^{-/-}$ MEFs was established based on previous reports[43]. U87MG and A172 cells were cultured in DMEM. All cells were supplemented with 10% heat-inactivated FBS, 50 IU/mL penicillin, and 50 μg/mL streptomycin in an incubator containing a humidified atmosphere of 95% air and 5% $CO_2$ at 37 °C. All the cells were checked quarterly for mycoplasma contamination using the first-generation MycoAlert™ mycoplasma detection kit (#LT07-118; Lonza). Erastin (10 mM), RSL3 (5 mM), and ML210 (5 mM) stock solution in DMSO were prepared and stored in a dark-colored bottle, from which desired dilutions were made. Cells were grown to about 70–80% confluence and then treated with specified drugs at required concentrations and for a different time period.

### Viability assays
To determine changes in cell viability[64]. Briefly, we plated 10000 cells/0.1 mL/ in 96-well flat-bottom plates and were subjected to RSL3 treatment. Subsequently, 25 μL of MTT (5 mg/mL) in PBS was added to each well at desired time intervals. The plates were incubated additionally for 2 h at 37 °C. After the incubation, the formazan crystals were solubilized in 200 μL of DMSO, and the absorbance at 570 nm was measured using an EnSpire™ multimode plate reader (PerkinElmer, Waltham, USA). For colony formation assay, $0.7 \times 10^6$ cells were seeded into a 3 mL plate to assess colony formation efficiency after being treated with RSL3 for the indicated time points. The colonies were fixed and stained with 0.25% crystal violet (dissolved in 50% methanol) for 30 min at 37 °C, followed by washing twice with PBS. Images were acquired using a phase-contrast microscope (Olympus, Tokyo, Japan) using a 20x objective.

## Protein lysate preparation and immunoblot analysis

Cells and tissues samples were lysed in ice-cold RIPA lysis buffer (50 mM Tris HCl - pH 7.4, 1% NP-40, 40 mM NaF, 10 mM NaCl, 10 mM Na3VO4, 1 mM PMSF, 10 mM DTT, and EDTA-free protease inhibitor tablet). The whole-cell lysates were centrifuged and quantified with Pierce BCA Protein assay Kit (#23225; Thermo Fisher, CA, USA). Lysates were mixed with 6X loading buffer and boiled at 100 °C for 3 min. Equal amounts of proteins were subjected to 10–15% SDS-PAGE, and the separated proteins were transferred to nitrocellulose membrane by wet transfer method using Bio-Rad electrotransfer apparatus. Following the transfer, blots were blocked with 5% non-fat milk in Tris-buffer saline containing 0.1% Tween-20 (TBST). Blots were then incubated with various primary antibodies. Following three washes in TBST, membranes were incubated with goat anti-mouse or anti-rabbit IgG HRP secondary at 37 °C for 2 h and washed. The antibodies used are shown in Supplementary Table 1. Protein signals were detected by using SuperSignal™ West Pico Plus (#34080; Thermo Fisher, CA, USA) or SuperSignal™ West Femto maximum sensitivity chemiluminescence substrate (#34095; Thermo Fisher, CA, USA), and blots were captured by Genegenome Imaging System (Syngene, MD, USA). The relative band intensity was quantified using the Image Studio Lite 5.2.5 software (LI-COR Biosciences).

## RNA isolation and quantitative PCR

RNA was isolated from cells using the RNeasy Mini kit (#74104; Qiagen, Valencia, CA, USA) according to the manufacturer's instructions. Quantitative real-time PCR was performed using Power SYBR Green® RNA-to-CT one-Step Kit (#4391178; Applied Biosystems, CA, USA), and the reaction products were run on StepOnePlus real-time PCR System (Applied Biosystems). Reactions were performed in triplicate according to the manufacturer's protocol. The data were normalized to the internal control GAPDH, and the fold change was calculated using the $2^{-\Delta\Delta ct}$ method. The primers synthesized and desalted from Integrated DNA Technology, USA, are listed in Supplementary Table 2.

## Intracellular ROS measurement

Intracellular ROS level was measured using an oxidation-sensitive fluorescent probe DCFH-DA. After treatment of cells with RSL3, cells were washed 2X with PBS and then treated with 25 μM of DCFH-DA at 37 °C for 30 min in the dark. Then, the cells were washed 3X with PBS to remove the extracellular dye, harvested with a policeman in a phenol red-free medium, and the DCF fluorescence was measured immediately using excitation (485 nm) and emission (535 nm) wavelengths using EnSpire™ multimode plate reader (PerkinElmer, Waltham, USA).

**GPX4 activity assay.** The glutathione peroxidase (GPx) activity was measured by using a colorimetric assay according to the manufacturer's protocol from Abcam (GPx Activity Kit; #ab102530). In brief, $2 \times 10^6$ cells were collected and depleted of all GSSG by incubating the sample with glutathione reductase (GR) and reduced glutathione (GSH) for 15 min. GPx activity was determined by adding cumene hydroperoxide and incubating for 0 and 5 min. The enzymatic reaction was run in 96-well plates, and oxidation of NADPH was determined by OD at 340 nm on an EnSpire™ multimode plate reader (PerkinElmer, Waltham, USA).

## Lipid peroxidation analysis by flow cytometry

Image-iT® kit was used to measure the lipid peroxidation through oxidation of the C11-Bodipy® 581/591 sensor[63]. Briefly, cells were treated with RSL3 or erastin for the indicated time points. After incubation, cells were washed with PBS, trypsinized, and pelletized by centrifugation. Then, the pellet was stained with C11-Bodipy 581/591 (2 μM) for 30 min at 37 °C. Oxidation of the polyunsaturated butadienyl portion of the dye resulted in a fluorescence emission peak shift from ∼ 590 nm to ∼ 510 nm. Data collection (10,000 cells/sample were gated on live cells by forward/side scatter and C11-Bodipy exclusion) was done immediately afterward on a BD FACSAriaIII or Canto II flow cytometry (Becton Dickinson, Heidelberg, Germany). The data was

analyzed using the BD FACSDiva software. Following this, the median fluorescence intensity (MFI) was quantified using FlowJo V.10.1 software.

## Detection of labile iron by imaging and flow cytometry

The Phen Green SK diacetate (PG SK) fluorophore sensor was utilized to detect the intracellular labile iron pool (LIP). PG SK is more sensitive to ferrous ($Fe^{2+}$) quenching than ferric ion ($Fe^{3+}$). Once it enters the cell, the diacetate group of PG SK is cleaved by intracellular esterases. PG SK then chelates LIP resulting in the quenching of its fluorescence[65]. Quenched fluorescence indicates increased LIP, whereas restored quenched fluorescence indicates decreased LIP. Intracellular LIP was further verified using the FerroOrange fluorescent probe, which enables live-cell fluorescence imaging of intracellular $Fe^{2+}$. Briefly, cells were seeded in six-well plates ($0.5 \times 10^6$ cells/well) one day before the experiment. The next day, cells were treated with RSL3 or erastin for the indicated time points. After incubation, cells were washed with PBS, trypsinized, and pelletized by centrifugation. Then, the pellet was stained with 500 μL of PG SK (5 μM) or FerroOrange (1 μM) in PBS and transferred to FACS tubes by incubating for 30 min at 37 °C in the dark. Data collection (10,000 cells/sample were gated on live cells by forward/side scatter and PGSK or FerroOrange exclusion) was done immediately afterward on a BD FACSAriaIII or Canto II flow cytometry (Becton Dickinson, Heidelberg, Germany). The data was analyzed using the BD FACSDiva software. Following this, the median fluorescence intensity (MFI) of PG SK or FerroOrange staining was then quantitated using FlowJo V.10.1 software. For imaging, cells plated on 6-well were treated with RSL3 for 3 h at 37 °C. Cells were then stained with 5 μM of PG SK for 30 min in a complete growth medium at 37 °C in the dark. For FerroOrange staining, cells were incubated with 1 μM of FerroOrange in a serum-free MEM medium. The cells were then washed 3X with PBS and then imaged under an IX73 inverted fluorescent microscope (Olympus, Tokyo, Japan) using a 20x objective with FITC or Texas Red channels.

## Plasmids and transient transfection

pCMV6-Myc-DDK-tagged PAR-4 (#RC202733) and pCMV Entry-Myc-DDK empty vector (#PS100001) were purchased from OriGene Technologies, Inc. (Rockville, MD, USA). In addition, dual fluorescent mRFP-EGFP-LC3 (ptfLC3) (Addgene plasmid # 21074) was a gift from Tamotsu Yoshimori. DNA transfection to cells was performed using MegaTran 1.0 transfection or HiPerFect transfection reagent according to the manufacturer's protocol.

## Lentivirus-mediated gene transfer and establishment of the stable U87MG cell line expressing Par-4 under doxycycline control

Lenti-Pac 293Ta cells (Genecopoeia) were co-transfected with lentiviral Dox-ON-Par-4-pEZ-Lv208 vector (EX-O0085-Lv208; Genecopoeia) together with packaging vectors using Lenti-Pac expression packing kit (Genecopoeia). At 48 h after transfection, a virus-containing medium was collected, supplemented with polybrene (4 mg/mL), and infected with U87MG target cells. The transduced cells were selected with puromycin (0.4 μg/mL) for 2 weeks to generate stable gene-expressing cell lines. Western blot analyses confirmed positive clones to express Par-4 protein.

## CRISPR/Cas9-mediated knockout of Par-4

CRISPR-Cas9 gene-editing technology was applied to generate stable Par-4 knockout in U87MG cells. Plasmid vectors, including pLV[2CRISPR]-hCas9:T2A:Puro-U6>hPAWR[sgRNA#281]-U6>hPAWR[gRNA#312] (#VB200608-1979enn) and scramble gRNA lentiviral control vector pLV[CRISPR]-hCas9/Puro-U6>Scramble_gRNA1(#VB010000-9355sqw), were purchased from VectorBuilder (Chicago, IL, USA). Par-4-specific single-guide RNA (sgRNA) was designed using the online tool CRISPR DESIGN (http://CRISPR.mit.edu) and was synthesized and cloned into Lenti-sgRNA-EGFR. The sgRNA and oligo sequences for constructing the CRISPR KO vector are shown in Supplementary Table 3. Lentiviruses were packaged in 293 T cells and then transfected into related cells.

Puromycin-selected and pooled stable cells were then subject to cellular assays and immunoblotting analysis.

## Lentiviral-mediated shRNA transduction of Par-4, ATG-5, and ATG-7

U87MG and A172 cells were transduced with lentiviral vectors carrying an shRNA to Par-4 (human Par-4 shRNA) lentiviral particle (#SC-36190-V) (Santa Cruz Biotechnology, CA, USA). In addition, scrambled control shRNA lentiviral particles (#SC-108080) (Santa Cruz Biotechnology) were used for comparison control. Briefly, cells were seeded into 12 well plates ($5 \times 10^4$ cells/well) and were grown to 50% confluence on the day of viral infection supplemented with polybrene (5 mg/mL). To generate a stable clone, the transduced cells were selected with puromycin dihydrochloride (0.4 mg/mL) for 2 weeks to develop a stable clone. Efficient Par-4 knockdown was confirmed by Western blot analysis. shRNAs for ATG-5 (TL314610), ATG-7 (TL314609), and shCont (TR30021) constructs in lentiviral GFP vector were purchased from OriGene Technologies (Rockville, MD, USA), and the specific target sequences are listed in Supplementary Table 4. According to the manufacturer's instructions, lentiviruses carrying the shRNAs were produced in 293 T cells using a Lenti-Pac expression packing kit (Genecopoeia, Rockville, MD, USA). U87MG cells were infected by these viruses and were selected with puromycin (0.3 µg/mL for ATG-5 and 0.1 µg/mL for the ATG-7 gene, respectively). Scrambled control shRNA was used as a control. The expression of the puromycin-resistant clones was identified using a Western blot.

## siRNA transfection

siPar-4 (#SC-36190), siAtg-5 (#SC-41445), siAtg-7 (#SC-41447), and siControl (#SC-37007) were purchased from Santa Cruz Biotechnology Inc. (Santa Cruz Biotechnology, CA, USA). For the transfection, cells were seeded into a six-well plate at a density of $0.2 \times 10^6$ cells per well and allowed to reach 50% confluence on the day of transfection. Cells were transfected with 30 nM siRNA using HiPerFect transfection reagent (Qiagen, Valencia, CA, USA) according to the manufacturer's protocol. The efficient knockdown of Par-4 was verified by Western blot analysis.

## Determination of autophagic flux

Tandem reporter constructs mRFP-GFP-LC3 (ptfLC3; Addgene plasmid # 21074) were used to monitor autophagic flux based on the pH stability of GFP and mRFP fluorescent proteins. GFP is quenched in acidic environments, while mRFP is relatively stable at the acidic pH found in lysosomes. Therefore, autophagosome formation leads to an increase of yellow puncta (GFP + /RFP + ), with these puncta turning red (GFP − /RFP + ) upon fusion of an autophagosome with a lysosome. An increase in autophagy leads to more yellow and red puncta, while a block at the fusion of autophagosomes with lysosomes leads to an increase in yellow puncta but a decrease in red puncta. U87MG cells transfected with ptfLC3 were pretreated with autophagy inhibitors in the presence or absence of RSL3 for 6 h. Following the treatment, fluorescence images were taken under Olympus IX73 fluorescent microscope. Cells treated with BafA1 as a negative control for autophagy.

## Xenograft tumor studies

Animal experiments were approved by the Institutional Animal Care and Use Committee (IACUC) of the New York University Abu Dhabi (NYUAD) (NYUAD IACUC 21-0005) and performed in accordance with the Guide for Care and Use of Laboratory Animals[66]. BALB/c nude mice (female, eight weeks old, 20 ± 0.7 g) were obtained from Jackson Laboratory, USA, and housed at the NYUAD Vivarium Facility under specific pathogen-free conditions. Mice were maintained in air-filtered cages with controlled temperature (20 ± 3 °C) and humidity (50%) in a 12 h light/dark cycle. Mice had access to standard chow and water ad libitum (Research Diets; New Brunswick, NJ). The control shRNA and Par-4 stable knockdown U87MG cancer cells were initially screened to check the presence of infectious agents and tested using the Cell Line Examination and Report

(CLEAR) PCR Panel at Charles River Laboratories, USA. The cell line screening is provided as non-GLP for research purposes only. The control shRNA and Par-4 knockdown U87MG cells (100 µL, $1 \times 10^7$ cells) were injected subcutaneously into the right flank of each mouse[67,68]. Mice were assessed daily for overt signs of toxicity. Tumor volume was determined by high-precision calipers (Thermo Fisher, USA) using the following formula: Tumor volume (mm³) = (tumor length) x (tumor width)²/2.

Once the tumor volume reached ~25 mm³, the mice were divided into four treatment groups (4 mice/group): 1. shRNAcont, 2. shRNAcont +RSL3, 3. Par-4-shRNA, and 4. Par-4-shRNA+RSL3. The mice were injected intraperitoneally (i.p) with saline (0.02% DMSO) and 4.4 mg/Kg of RSL3 per animal in 100 µL, respectively, three times per week for 21 days[52–55]. When saline and RSL3 were injected, the tumor size and mice weight were measured using a digital caliper and a digital calculation balance (Magnusson Stainless Steel Metric Digital Caliper (23 cm) were obtained from Screwfix Direct Limited, Somerset, UK). At the endpoint, related mice were sacrificed, and the final tumor volume and mass were measured. Subsequently, the vital organs such as heart, lungs, liver, spleen, kidney, and tumor tissues were fixed in 10% neutral buffered formalin, followed by paraffin-embedding for H&E analysis, and the tissue sections were dewaxed and stained using standard procedures[67,68] or snap-frozen in liquid nitrogen then stored at -80 °C until immunoblotting investigation.

## RNA-Seq

**RNA isolation and purification.** The total RNA was extracted from U87MG ($1 \times 10^6$ cells in a 3 mL plate) treated with vehicle (control, $n = 3$), RSL3 (2 µM, $n = 3$) using the RNeasy Mini kit (#74104; Qiagen, Valencia, CA, USA), according to the manufacturer's instructions. For the QC of purified RNA, absorbance ratios were assessed with NanoDrop 2000 spectrophotometer (Thermo Fisher Scientific, Waltham, MA).

**RNA-Seq library preparation, sequencing, and processing.** Total RNA quality was evaluated based on the RNA integrity number (RIN) using a BioAnalyzer 2100 (Agilent Technologies, Inc., Santa Clara, CA, USA) and quantified with a Qubit 4 Fluorometer. Next, RNA-Seq libraries were prepared using the NEB Next Ultra II RNA Library Prep Kit (#E7770L; New England Biolab, Ipswich, MA, USA), following the manufacturer's recommendations. Total RNA samples (250 ng) were subjected to cDNA construction for Illumina sequencing following the mRNA-Seq sample preparation kit protocol. Oligo(dT) magnetic beads were used to isolate poly(A) RNA from the total RNA samples. The mRNA was fragmented, and the first-strand cDNA was reverse-transcribed using random primers, followed by second-strand cDNA synthesis. After being end-repaired and A-tailed, the resulting double-stranded cDNA was ligated to NEBNext® adapters. Finally, PCR amplification of 12 cycles was done for enrichment, producing a 350–400 bp fragment, including adapters. The fragment size and purity of the libraries were assessed on a BioAnalyzer 2100 instrument (Agilent Technologies, Inc., Santa Clara, CA, USA). Quantifying the pooled libraries required for RNA-seq was determined by real-time qPCR using a KAPA's library quantification kit for the Illumina sequencing platforms (#KK4835; Kapa Biosystems, Inc. Wilmington, MA, USA). Pooled libraries were sequenced using the Illumina NextSeq 500/550 System with a Paired-end Mid Output v2 kit, 150 cycles (Illumina, San Diego, CA, USA). For analysis, reference human genome and gene information were downloaded from the National Center for Biotechnology Information (NCBI) database. Raw reads were filtered to produce high-quality clean data. All the subsequent analyses were performed with clean data.

**Bioinformatic and computational analysis.** Raw FASTQ sequenced reads were first assessed for quality using FastQC v0.11.5. The reads were then subjected to Trimmomatic v0.36 for quality trimming and adapter sequence removal, with the parameters (ILLUMINACLIP: trimmomatic_adapter.fa:2:30:10 TRAILING:3 LEADING:3 SLI-DINGWINDOW:4:15 MINLEN:36)[69]. The surviving trimmed read

pairs were then processed with Fastp for poly-G tails and Novaseq/Nextseq-specific artefacts[70]. After the quality trimming, the reads were assessed again with FastQC v0.11.5.

Subsequently, the reads were aligned to the human reference genome GRCh38.p4 using HISAT2 with the default parameters and an additional –dta flag[71]. The SAM alignments were then converted to BAM format and sorted coordinate using SAMtools v1.3.1[72]. The sorted BAM files were then passed through HTSeq-count v0.6.1p1 using the following options (-s no -t exon -I gene_id) for raw count generation[73]. Concurrently, the sorted alignments were processed with Stringtie v1.3.0 for transcriptome quantification[74]. Briefly, the process looks like this, stringtie -> stringtie merge (to create a merged transcriptome GTF file of all the samples) -> stringtie (this time using the GTF generated by the previous merging step). Finally, Qualimap v2.2.2 was used to generate RNAseq-specific QC metrics per sample[75]. Raw reads count from each sample were used to identify differentially expressed genes between RSL3 and control using DESeq2 using deployed under the NASQAR, a web-based platform[76]. Surrogate variable analysis was conducted for hidden batch-effect correction before inputting the data into DESeq2. Bioinfo-Kit and pheatmap packages generated volcano plots and heatmaps, respectively. Heatmaps or bar graphs were generated ($P < 0.05$) with the selected significant DEGS associated with ferroptosis-related cell death pathways.

**Statistics and reproducibility**. For knockout or knockdown genes of interest, at least two independent sgRNAs or shRNAs were employed to generate cell lines, and similar were found in all cell lines. For over-expression experiments, lentivirus containing the gene of interest was prepared multiple times to verify overexpression efficiency in respective cell lines. All statistical analysis was performed using GraphPad Prism 10.1.1 (GraphPad Software, San Diego, CA, USA). The results are presented as the mean ± SD of three biologically independent experiments or samples. Statistical significance ($P$ values) was analyzed by one-way ANOVA using the Bonferroni post-hoc test. For in vivo studies, the investigators were blinded in certain parts of the experiment, including work design, treatment, data acquisition, and data analysis. Another investigator carried out these parts. A power calculation was used for the animal studies to select sample sizes from the New York University Abu Dhabi Institutional Animal Care and Use Committee (NYUAD-IACUC) Protocol (Protocol No. 21-0005). For the in vivo studies, statistical analysis was performed using Two-way ANOVA followed by Tukey's post hoc test between control and treatment groups. The difference was considered significant for all the statistical analyses with a $P$-value less than 0.05. The experiments were repeated three times for immunoblots with similar results, and representative data were shown.

### Reporting summary
Further information on research design is available in the Nature Portfolio Reporting Summary linked to this article.

### Data availability
The RNA-Seq data were deposited into Gene Expression Omnibus under accession number GSE267823. The original uncropped western blots for Figs. 1–6 and Supplementary Figs. 1–6, along with the numerical source data behind the graphs in the paper, can be found in Supplementary Data 2. All other data are available from the corresponding author on reasonable request.

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

## Acknowledgements

This research received financial support from the ASPIRE Award for Research Excellence (Grant No. S1158) and partially from NYUAD research grant AD252 awarded to S.G. We also acknowledge the assistance of the NYUAD Core Technology Platform. Graphics figures were created using BioRender.com.

## Author contributions

K.S. conceptualized and designed the project. K.S., S.G., F.T., A.R.C., S.P., and A.A. designed and performed the experiments and data analyses. S.G. supervised the project. V.M.R. made valuable suggestions in designing the study and constructed the Par-4$^{+/+}$ and Par-4$^{-/-}$ knockout MEFs. K.S., F.T., and S.P. generated stable knockdown cell lines. A.R.C. assisted in MEFs-related experiments. M.S. performed RNA sequencing work and assisted with FACS analysis. N.D. and M.A. performed bioinformatic analysis, and K.S. and S.P. assisted with the bioinformatic data interpretations. M.M. supervised animal work, and P.L. designed and performed animal experiments. K.S. wrote the manuscript with contributions from S.G., F.T., A.R.C., S.P., A.A., V.M.R., P.L., and M.M. All authors approved and commented on the manuscript.

## Competing interests

The authors declare no competing interests.
