## [Peer Review File · Communications Biology]

Reviewers' comments:

Reviewer #1 (Remarks to the Author):

Here the authors present a study demonstrating that Par-4 activation sensitizes cells to ferroptosis via upregulation of ferritinophagy. The study is of interest but there are a number of issues that require attention:

1. Figure 1a - What was the effect of 2 μ m RSL3 treatment for 24h on the cells? Some supplementary data should be presented regarding this experiment. What was the IC50 of RSL3 in these cells? Are the cells all dead / dying at 24h with this level of RSL3 or not? The data related to this question appear to come up later in the figure (1i) but this would seem to be critical information to interpret the RNA-Seq experiment and the initial finding about Par4. RSL3 targets GPX4 but is viewed as having many off-target effects. Have the authors tried more specific GPX4 targeting agents such as ML-210 or JKE-1674. Demonstrating similar effects of JKE-1674 on FTH1 / NCOA4 would further support their findings.

Second: What cut-offs did the authors use of determining differentially expressed genes. It appears to be just a significance cut-off and not a log 2 fold change as well. A more reasonable approach would be to set both as thresholds. This is mentioned as NCOA4 and GPX4 may be 'significantly' different but the log2 fold change of -0.214036984 may not be biologically significant.

2. Figure 1b. What do the colors correspond to in the heat map? It appears all identified hits are presented as either red or green and no gradations in between. The NCOA4 change reported in the supplemental table is -0.21 which would be darker green according to the heat map bar. The range chosen for the heat map is quite constrained - again calling into question how biologically significant these changes are. Furthermore there is no need to present the control row as being opposite the RSL3 condition as if this is log2GC then it is the ratio of RSL3 to Control.

3. 1c is this relative mRNA levels? If so, please note in the y-axis

4. Figure 1d and every other western blot in the manuscript: all western blots appear to have been adjusted using a high contrast setting. This presents a problem as overexposure may mask additional bands both non-specific and relevant ones (e.g. see Figure 5f - FTH1 blot - there is a lower band that is likely a lysosomal degradation product of FTH1 that has been obscured due to the high contrast used). Given this issue is present throughout every single western in the manuscript, the original unaltered blots should be presented in the supplementary data for evaluation.

5. Figure 2a - it's not clear what is meant by Par4 inhibited 'RSL3- and erosion-induced GPX4' The western levels are presented as a measure of activity. Whereas glutathione peroxidase activity is not evaluated. Similarly in Fig. 2e (although here it is described as GPX4 degradation as opposed to activity)

6. Figure 2b PG SK is not specific for iron. More specific dyes include FerroOrange. A control with DFO should be presented as well.

7. Figure 2l: how does the level of Par4 compare to endogenous levels of Par4?

8. Fig. 3n, o as in previous figures: to determine that Par4 expression is actually increasing autophagic flux a western based experiment would require +/- BafA1 conditions

9. The statement "Induction of ferritinophagy triggers labile iron accumulation, ROS generation, lipid peroxidation, and cell death^{7,37,47,48}." Is not fully supported by these citations with respect to the cell death part. More accurate would be that induction of ferritinophagy increases the sensitivity to ferroptosis inducers and thereby cell death under conditions of ferroptosis induction.

10. As it relates to the data for Par4 controlling NCOA4 / FTH1 degradation: in addition to a western blot, an evaluation of localization is warranted. Specifically is FTH1 no longer trafficking to the lysosome in Par4 KO or KDs?

11, Figure 4b - the 100 um DFO condition alone demonstrates a steady level of FTH1 if not increased. This is inconsistent with the known role of DFO to induce iron starvation and induce FTH1 turnover (decrease in FTH1 band expected). The methods should be updated to indicate the amount of time DFO is applied to the cell. If for a very short time, DFO may not have time to decrease iron levels sufficiently to induce ferritinophagy.

12. Figure 6 - RSL3 is not typically used for IP dosing. Rather RSL3 is used as an intratumoral injection.

Reviewer #2 (Remarks to the Author):

In this manuscript, the authors found that prostate apoptosis response-4 (Par-4) promotes ferroptosis possibly through activation of ferritinophagy (autophagic degradation of ferritin) via the nuclear receptor co-activator 4 (NCOA4). The role of NCOA4-mediated ferritinophagy has been clarified in the process of ferroptotic cell death. This study linked Par-4 to NCOA-mediated ferritinophagy in the regulation of ferroptosis. Though the finding expands our knowledge in the understanding of iron metabolism during ferroptosis, there are several issues needed to be addressed before consideration for publication.

Major points:

1. The rationale to focus the study of Par-4 in the regulation of ferroptosis is not strong enough. Why the authors specifically pick Par-4 as a candidate to start? Because there are so many genes changed during RSL3 treatment, any of them could be a key mediator or regulator of ferroptosis.

More evidence should be provided to support the logic. Meanwhile, it's hard to find Par-4 in Fig.1 A and B. Besides, RSL3 is known to induce autophagy. Is it possible that the induction of Par-4 by RSL3 is simply due to autophagy induction?

2. Whether another type of ferroptosis inhibitor (iron chelator) could suppress Par-4 expression induced by ferroptosis inducer ?

3. Fig.1i-k just showed these cells could response for ferroptosis induction, but what's the point to show it here?

4. Fig.2a showed that knockdown of Par-4 upregulated GPX4, which is known to suppress ferroptosis. The author need to prove the role of GPX4 in the Par-4 regulation of ferroptosis.

5. Erastin is reported to suppress GPX4 protein level through inhibiting mTORC1 signaling and GPX4 translation. While the authors showed KO of Par-4 restored GPX4 level under erastin treatment.

How to explain this if GPX4 proteins translation is already blocked?

6. Free labile iron is known to promote ferroptosis. In fig2g KO of Par-4 increased iron level, which seems contradict to the role of Par-4 in the regulation of ferritinophagy.

7. If Par-4 regulate NCOA4, which KO of Par-4 didn't upregulate NCOA4 in Fig. 4 ?

8. More evidence are needed to demonstrate the role of NCOA4 in the Par4 regulation of ferroptosis.

9. RSL3 treatment-induced ROS/lipid ROS increased Par-4 and autophagy, how about the regulation of Par-4 by other oxidative stress conditions?

10. In Fig. 6, the authors used RSL3 to induce ferroptosis in vivo, which is not appropriate. IKE is the only ferroptosis inducer suitable for in vivo study.

Reviewer #3 (Remarks to the Author):

In this study, the authors demonstrated the activation of a multi-faceted tumor-suppressor protein postate apoptosis response-4, Par-4/PAWR in glioblastoma (GBM) U87MG cells treated with ferroptosis activator 1S,3R-Ras Selective Lethal 3 (RSL3). Functional studies reveal that genetic depletion of Par-4 effectively blocks ferroptosis, whereas Par-4 overexpression sensitizes cells to undergo ferroptosis. The similar experimental works were also conducted in Par-4 wild type and Par-4 knockout primary mouse embryonic fibroblasts (MEFs). Thus, the authors claimed that Par-4, as a tumor suppressor, may play a vital and novel role in ferroptotic cell death upon ferroptosis stimulation. The manuscript seemed to be well documented and experiments also seemed to be well-done.

I have several general suggestions that the authors must have thought of, based on the data presented here. Both of them would make this paper of relevance for a broader audience.

1) Abstract section, the authors claimed that they used an unbiased genome-wide screening to identify Par-4. Actually, the authors provided the RNA sequencing (RNA-seq) analysis. Please keep

this consistence.

2) It has been noticed that Par-4 induces the activation of ferritinophagy via NCOA4. In this case, it is not necessary to confirm the former findings.

3) It has been shown that ferroptosis is also an autophagy-dependent cell death. Upon autophagy inducer treatment (such as trehalose or metformin), does Par-4 enhance its tumour suppressor function?

4) Page 169-171, "...the expression of p53 over time in U87MG cells (Supplementary Fig. 2j), suggesting that Par-4-mediated ferroptosis induced by RSL3 is independent of p53 expression." This statement may be less conclusive. Since the regulatory role of autophagy by p53 functions diversely on its cellular localization, the author would perform the functional assay of p53, such as the localization change of p53.

Detailed Point-by-point response to the reviewer's comments

Note to Reviewers: We extend our gratitude to all the reviewers who have taken the time to review our manuscript and provided us with valuable feedback. We have carefully considered all the comments made by the reviewers and provided a detailed point-by-point response to address them. To make it easier for the reviewers to follow our rebuttal letter and revised manuscript, we have presented all the new data as rebuttal figures in this letter, with references to the corresponding figures and text in our revised manuscript. We have highlighted in yellow all the changes made in the revised manuscript. To ensure clarity, we have quoted the reviewers' comments verbatim in plain text and provided our responses in plain blue text.

Please note that we have updated the title of the revised manuscript. The original title submitted was "Par-4 is a novel regulator of selective autophagy-driven ferroptosis," but we have changed it to "**Tumor suppressor Par-4 activates autophagy-dependent ferroptosis.**" This was done to comply with the journal's policy of avoiding hyperbolic words such as "novel." We believe that this new title accurately reflects the manuscript's focus and content.

Reviewer #1 (Reviewer Comments to the Author):

Here the authors present a study demonstrating that Par-4 activation sensitizes cells to ferroptosis via upregulation of ferritinophagy. The study is of interest but there are a number of issues that require attention:

1. Figure 1a - What was the effect of 2 μM RSL3 treatment for 24h on the cells? Some supplementary data should be presented regarding this experiment. What was the IC50 of RSL3 in these cells? Are the cells all dead / dying at 24h with this level of RSL3 or not? The data related to this question appear to come up later in the figure (1i) but this would seem to be critical information to interpret the RNA-Seq experiment and the initial finding about Par4. RSL3 targets GPX4 but is viewed as having many off-target effects. Have the authors tried more specific GPX4 targeting agents such as ML-210 or JKE-1674. Demonstrating similar effects of JKE-1674 on FTH1 / NCOA4 would further support their findings.

Response: Firstly, we followed this suggestion and carried out the dose-responsive effect of RSL3 for 24 h in U87MG cells (the cell line extensively used in this study). As shown in the **Rebuttal Fig. 1**, we demonstrate that RSL3 induces cell death in a dose-dependent manner. RSL3 decreased cell viability substantially at low concentrations (~75 and 90% cell death at 0.5 and 1 μM RSL3, respectively), and the viability remained relatively unchanged at higher concentrations (i.e. > 1 μM RSL3). We have added this result in the **NEW Supplementary Fig. 1b** of the revised manuscript.

Rebuttal Figure 1: Dose-responsive effect of RSL3 in U87MG cells. U87MG cells were treated with the indicated concentration of RSL3 for 24 h. Cell viability was analyzed by MTT assay. Data shown are mean \pm SD; n = 3 samples.

The data related to this question appear to come up later in the figure (1i) but this would seem to be critical information to interpret the RNA-Seq experiment and the initial finding about Par4.

Regarding the additional question raised by the reviewer about Figure 1i-k cell viability (now moved to **Supplementary Fig. 1g-i** in the revised manuscript), it is important to note that ferroptosis was first discovered during the characterization of cell death induced by RSL3 and erastin. These compounds were later categorized as class 1 and 2 ferroptosis inducers [1,2]. In any case, we further confirmed that in glioma cells (U87MG and A172), erastin- or RSL3-induced cell death can be rescued by the ferroptosis inhibitors (ferrostatin-1 and liproxstatin-1) and iron chelator deferoxamine (DFO), but not by other cell death inhibitors (apoptosis inhibitor Z-VAD-FMK and necroptosis inhibitors such as GSK-963, GSK-872, and necrosulfonamide [NSA]). This demonstrated that erastin or RSL3 induces ferroptosis, but not other forms of regulated cell death, in U87MG and A172 cells (**Supplementary Fig. 1d-i**). Part of this discussion has been included in the results section of the revised manuscript (see **page 6** and **line numbers 134-139**). Furthermore, to avoid confusion, Figure 1i-k has been relocated to **Supplementary Fig. 1g-i** in the revised manuscript.

We also provided additional clarification regarding the RNA sequencing experiments depicted in Fig. 1a and b of the original manuscript. In this study, we observed the activation of Par-4, which is the target of interest, during ferroptosis in response to RSL3 treatment for 24 h. We induced ferroptosis in U87MG glioma cells using RSL3 at 2 μ M for 24 h, and then RNA sequencing was performed on the treated and non-treated cells. We found that the mRNA level of Par-4/*PAWR* was significantly up-regulated in the RSL3-treated U87MG cells compared to the non-treated group (**Fig. 1a volcano and 1b bar plot** in the revised manuscript). We used qPCR and western blot to corroborate these results to detect Par-4 mRNA and protein expression under different treatments in glioma cells. Consistent with the RNA sequencing results, the Par-4 mRNA level was up-regulated after the addition of RSL3 in U87MG cells compared to the control group. Similarly, results obtained at the protein level were consistent, indicating that Par-4 may participate in the process of ferroptosis (**Fig. 1c and d** in the revised manuscript). We hope this explanation helps clear up any confusion the reviewer may have had regarding **Fig. 1a and b**.

RSL3 targets GPX4 but is viewed as having many off-target effects. Have the authors tried more specific GPX4 targeting agents such as ML-210 or JKE-1674. Demonstrating similar effects of JKE-1674 on FTH1 / NCOA4 would further support their findings.

The reviewer has suggested using more specific GPX4 targeting agents like ML210 or JKE-1674 to strengthen our findings. ML210 is a commonly used ferroptosis-inducing compound that directly inhibits the GPX4. As suggested by the reviewer, we investigated whether Par-4 could be activated by ML210 during ferroptosis. We treated U87MG cells with ML210 in different concentrations for 3 h and presented the new data in **Rebuttal Fig. 2a** and **NEW Supplementary Fig. 1a** of the revised manuscript. Like RSL3, treatment with ML210 induced concentration-dependent activation of Par-4 in U87MG cells, indicating that Par-4 is activated during ferroptosis, irrespective of the ferroptosis inducers. Additionally, we have strengthened several of our original conclusions further by repeating those experiments using ML210. For instance, we have conducted an experiment to verify that Par-4 fosters activation of autophagy signaling *via* NCOA4-mediated ferritinophagy. Our Western blot analysis showed that genetic inhibition of Par-4 (siPar-4) significantly reversed the degradation of GPX4, conversion of LC3-II, and reduction of p62/SQSTM1, FTH1, and NCOA4 in glioma cells induced by ML210 (**Rebuttal Fig. 2b and c** and **NEW Supplementary Fig. 2d and 3j**). The same results were also observed in double knockout Par-4^{-/-} MEFs, in which the absence of Par-4 significantly inhibited ML210-induced GPX4, FTH1, and NCOA4 degradation and prevented ML210-induced cell death (**Rebuttal Fig. 2d-f** and **NEW Supplementary Fig. 2h, 2l and 4e**). These data support the conclusion that Par-4 regulates ferroptosis irrespective of the ferroptosis inducers used. Although out of the scope of the current study, we plan to provide more detailed mechanistic insights into this regulation in our follow-up studies.

Rebuttal Figure 2: Par-4 activation is essential for ferroptosis.

(a) U87MG cells were treated with the indicated concentration of ML210 for 3 h, and then, a Western blot analysis of Par-4 was determined. (b and c) U87MG cells were transiently transfected with siPar-4. After transfection, cells were treated with ML210 (2.5 μM) for 3 h. Following the treatment, a Western blot analysis of indicated proteins was performed. Par-4^{+/+} MEFs and Par-4^{-/-} MEFs were also treated with ML210 (2.5 μM) for 3 h. After the treatment, (d and e) Western blot analysis indicated proteins, and (f) cell viability was analyzed. The relative density of protein bands was quantified and normalized to the actin of each group, and fold changes were presented in histograms from three independent experiments. Data shown are mean ± SD; n = 3 samples.

Second: What cut-offs did the authors use of determining differentially expressed genes. It appears to be just a significance cut-off and not a log 2 fold change as well. A more reasonable approach would be to set both as thresholds. This is mentioned as NCOA4 and GPX4 may be 'significantly' different but the log2 fold change of -0.214036984 may not be biologically significant.

2. Figure 1b. What do the colors correspond to in the heat map? It appears all identified hits are presented as either red or green and no gradations in between. The NCOA4 change reported in the supplemental table is -0.21 which would be darker green according to the heat map bar. The range chosen for the heat map is quite constrained – again calling into question how biologically significant these changes are. Furthermore there is no need to present the control row as being opposite the RSL3 condition as if this is log2GC then it is the ratio of RSL3 to Control.

Response: We would like to address questions 1 and 2 together since they both relate to Fig. 1b. We apologize for any confusion caused by the analysis of differentially expressed genes (DEGs) in Fig. 1b. To identify the DEGs, we used a statistical significance cut-off of $\text{padj} < 0.05$, along with gene expression changes measured in log2fold change. For instance, you can find the corresponding log2fold changes and padj values for the gene names in **Supplementary Table 1**.

Regarding your example, we can confirm that NCOA4 and GPX4 are indeed significantly downregulated in RSL3 when compared to the controls, and the -0.214036984 and -0.397884668 values represent the log2fold changes in the expression of these genes. Although the change in

expression may not be substantial, it is statistically significant in RSL3 samples when compared to the controls (n=3). This consistency is maintained throughout **Supplementary Table 1**.

We agree with the reviewer's comment regarding the control row being opposite to the RSL3 condition in the log2fold change in Fig. 1b. To address this concern, we have replaced the **Fig. 1b** heatmap with a bar plot (**Rebuttal Fig. 3**) in the revised manuscript. The new figure now demonstrates the log2-fold change in expression of the ferroptosis-related genes following RSL3 treatment compared to control U87MG cells. All the identified hits are appropriately color-coded for their corresponding genes without the control, as suggested by the reviewer. Significantly upregulated and downregulated genes are depicted in red and green colors, respectively. This information has now been added to the updated Fig. 1 legend of the revised manuscript. We hope this clarification clears the confusion in **Fig. 1b**.

Rebuttal Figure 3: Identification of Par-4/PAWR as a critical contributor to ferroptosis: Bar plot demonstrates the log2-fold change in expression of the ferroptosis-related genes following RSL3 treatment compared with control U87MG cells (n= 3 samples). The selected genes were identified as related to ferroptosis using FerrDb at <http://www.zhounan.org/ferrdb/>. The bars are labeled with gene symbols. Significantly upregulated and downregulated genes are depicted in red and green, respectively. The gene of interest, *PAWR* (Par-4), is highlighted by the red arrow in Fig. 1b.

3. 1c is this relative mRNA levels? If so, please note in the y-axis

Response: Thank you for your helpful suggestion. We have revised the manuscript by changing "Par-4 mRNA" to "Relative mRNA levels" on the y-axis of Fig. 1c. This information has now been added to the updated Fig. 1c legend section.

4. Figure 1d and every other western blot in the manuscript: all western blots appear to have been adjusted using a high contrast setting. This presents a problem as overexposure may mask

additional bands both non-specific and relevant ones (e.g. see Figure 5f - FTH1 blot - there is a lower band that is likely a lysosomal degradation product of FTH1 that has been obscured due to the high contrast used). Given this issue is present throughout every single western in the manuscript, the original unaltered blots should be presented in the supplementary data for evaluation.

Response: Thank you for your helpful suggestion. We have submitted an Excel file named "Source Data" that contains all original, unaltered Western blot images for the main and supplementary figures.

5. Figure 2a - it's not clear what is meant by Par4 inhibited 'RSL3- and erastin-induced GPX4' The western levels are presented as a measure of activity. Whereas glutathione peroxidase activity is not evaluated. Similarly in Fig. 2e (although here it is described as GPX4 degradation as opposed to activity)

Response: Glutathione peroxidase 4 (GPX4) is the critical enzyme that can reduce lipid peroxidation and prevent ferroptosis [2]. In our manuscript, we demonstrated that RSL3 and erastin (the most commonly used ferroptosis inducers) [2] cause Par-4-dependent GPX4 degradation, leading to ferroptosis (Fig. 2a of the revised manuscript). As a confirmation, we genetically knocked-down Par-4 and tested if that would suppress GPX4 degradation. Our results indicate that the genetic deletion of Par-4 through shRNA, CRISPR, or siRNA significantly reduced the RSL3- and erastin-mediated GPX4 degradation (**as shown in Fig. 2a and e** in our revised manuscript). Similar results were obtained using homozygous knockout Par-4^{-/-} primary mouse embryonic fibroblasts (MEFs) (**as shown in Fig. 2f** in our revised manuscript). These findings highlight the significance of Par-4 as a novel and critical regulator of ferroptosis.

We apologize for any confusion caused by Fig. 2a, 2e, and 2f in the western blot experiments, as indicated by the reviewer. Kindly note that our western levels in Fig. 2a, 2e, and 2f are not presented as a measure of activity but rather presented as the GPX4 protein level (fold change relative to control). We provided a clear explanation of the western blot quantification in the corresponding figure legends in the revised manuscript. We hope this clarification clears the confusion.

Thanks to the reviewer, we have also considered measuring the GPX4 activity to reinforce our hypothesis. As shown in the **Rebuttal Fig. 4 (NEW Supplementary Fig. 2a)** in the revised manuscript, genetic suppression of Par-4 significantly increased GPX4 activity in A172 cells in response to erastin, compared to the control shRNA group. Interestingly, the knockdown of Par-4 alone had higher GPX4 activity than the control shRNA group. Altogether, these findings suggest that Par-4 inhibits GPX4 activity by promoting GPX4 protein degradation, which leads to lipid hydroperoxide accumulation and ferroptosis. This information has now been updated in the result section of the revised manuscript (**Supplementary Fig. 2a**). The figure legends and methods section has been revised accordingly (**Pages 21-22**).

Rebuttal Figure 4: Par-4 inhibits GPX4 activity during ferroptosis. A172 cells were stably transfected with Par-4-shRNA. After transfection, cells were treated with erastin (10 μ M) for 24 h. Following the treatment, GPX4 activity was measured. Data shown are mean \pm SD; n= 3 samples.

The relevant section of the **Results** has been modified accordingly and now reads as follows:

Par-4 activation is critical for ferroptosis in human glioblastoma cells.

The suppression of Par-4 significantly inhibited both RSL3- and erastin-induced GPX4 protein degradation (Fig. 2a). Subsequently, we determined whether Par-4 knockdown affects the GPX4 activity. The suppression of Par-4 significantly increased GPX4 activity in erastin-treated A172 cells. Interestingly, Par-4 knockdown alone had higher GPX4 activity than the control shRNA group (Supplementary Fig. 2a). Moreover, Par-4 inhibition also prevented labile iron accumulation (Fig. 2b and Supplementary Fig. 2b), and lipid peroxidation (Fig. 2c), which are surrogate markers for ferroptosis. Additionally, suppression of Par-4 significantly prevented both RSL3- and erastin-induced cell death (Fig. 2d).

6. Figure 2b PG SK is not specific for iron. More specific dyes include FerroOrange. A control with DFO should be presented as well.

Response: According to this suggestion, we presented new data in **Fig. 4d and e** of the revised manuscript and the **Rebuttal Fig. 5**. We used FerroOrange dye, a fluorescent probe, to detect the free labile iron pool (LIP) in RSL3-treated cells. Following the treatment, fluorescence intensity was recorded under a fluorescence microscope and was further quantified using flow cytometry. As shown in **Rebuttal Fig. 5a** and **NEW Fig. 4d** in the revised manuscript, RSL3 treatment increased LIP levels, as seen by the red-orange fluorescence accumulated in the cells. However, pre-treatment with the iron chelators ciclopirox (CPX) and deferoxamine (DFO) blocked the red-orange fluorescence, as expected. The FerroOrange median fluorescence intensity (MFI) was further quantified using flow cytometry (**Rebuttal Fig. 5b** and **NEW Fig. 4e** in the revised manuscript). Notably, this result aligns with results from Phen GreenTM SK, a fluorescent indicator used to assess LIP levels in the original manuscript. This information has now been updated in the result section of the revised manuscript (**Fig. 4d and e**). Figure legends and method sections are revised accordingly (refer to **pages 22-23** under "Detection of labile iron by imaging and flow cytometry").

The relevant section of the **Results** has been modified accordingly and now reads as follows:

Par-4 regulates ferroptosis by inducing ferritinophagy. Similar results were also observed with FerroOrange dye, a fluorescent probe, to detect the intracellular labile iron levels. RSL3 treatment increased intracellular labile iron levels, as seen by the red-orange fluorescence accumulated in the cells. Pre-treatment with the iron chelators ciclopirox (CPX) and DFO blocked the red-orange fluorescence, as expected (Fig. 4d). The FerroOrange median fluorescence intensity (MFI) was further quantified using flow cytometry (Fig. 4e), suggesting the alterations in ferritin and NCOA4 level increase the labile iron pool *via* ferritinophagy in glioma cells.

In addition to this, we have used FerroOrange to examine whether suppressing the expression of Par-4 affects the accumulation of free labile iron pool (LIP). Our results showed that homozygous knockout Par-4^{-/-} MEFs significantly reduced RSL3-induced LIP when compared to Par-4^{+/+} MEFs (**Rebuttal Fig. 5c** and **NEW Supplementary Fig. 2j**). This finding provides further evidence to support the proposed critical role of Par-4 activation in ferroptosis through iron. In addition, the FerroOrange median fluorescence intensity (MFI) was confirmed by flow cytometry, as shown in **Rebuttal Fig. 5d** and **NEW Supplementary Fig. 2k**. We have updated the Supplementary Fig. 2 and 4 legend panel to include this information and provided more detailed information in the methods (refer to pages 23-24 under "Detection of labile iron by imaging and flow cytometry").

Rebuttal Figure 5: Par-4 promotes iron accumulation during ferroptosis. A172 cells were treated with RSL3 (2 μM) for 3 h in the presence or absence of CPX (10 μM) or DFO (100 μM). Similarly, Par-4^{+/+} MEFs and Par-4^{-/-} MEFs were also treated with RSL3 (2 μM) for 3 h. After the treatment, the cells were subjected to FerroOrange staining to evaluate intracellular LIP (a and c). Red fluorescence signals were captured and visualized through a fluorescent microscope using constant fluorescence parameters described in the methods section: scale bar, 50 μm. Subsequently, fluorescence intensity was quantified by flow cytometry analysis. The bar graph shows relative levels of LIP by FerroOrange staining in the indicated cells (b and d). Data shown are mean ± SD; n = 3 samples.

7. Figure 2l: how does the level of Par4 compare to endogenous levels of Par4?

Response: We appreciate the reviewer for asking this question. We present Fig. 2l data as **Rebuttal Fig. 6** to answer the reviewer's question. In this particular experiment, U87MG cells transiently transfected with Par-4 endogenous human pCMV6-DDK vector were treated with RSL3 (2 μM) for 3 h. The overexpression of Par-4 was confirmed by using an anti-DDK antibody. In our experiments, we found that the levels of Par-4 in overexpressed cells were significantly higher than the endogenous Par-4 level (**Rebuttal Fig. 6, lane 3**). Interestingly, our results indicate that adding endogenous Par-4 is sufficient to induce ferroptosis, as evidenced by basal levels of GPX4 being further decreased with or without RSL3 treatment. Our current findings are consistent with other reports that endogenous Par-4 is crucial for inducing apoptosis [3,4]. We have integrated these discussions into the revised manuscript (see **pages 15-16**).

Rebuttal Figure 6: Par-4 alone is sufficient for inducing ferroptosis. U87MG cells were transiently transfected with either 2 μ g of the empty vector or the Par-4 endogenous human pCMV6-DDK vector. After the transfection, cells were treated with RSL3 2 μ M, and whole-cell lysates were immunoblotted to detect DDK-Par-4, Par-4, and GPX4. Actin was used as a loading control.

8. Fig. 3n, o as in previous figures: to determine that Par4 expression is actually increasing autophagic flux a western based experiment would require +/- BafA1 conditions

Response: We tested whether endogenous Par-4 increases autophagic flux following the treatment of Bafilomycin A1 (BafA1). BafA1 can interfere with autophagosomes and prevent the degradation of autophagic proteins such as LC3-II and p62/SQSTM1. This means that if the amount of LC3-II and p62/SQSTM1 increases after treatment with bafilomycin A1, it indicates autophagic degradation, also known as "autophagic flux." As observed from the data presented below, LC3-II and p62/SQSTM1 levels were increased in empty vector cells treated with BafA1 alone. However, when RSL3 was added to these cells, the levels of LC3-II and p62/SQSTM1 were further increased, indicating that RSL3 induces increased autophagic flux (see **Rebuttal Fig. 7 and NEW Supplementary Fig. 3m**). Interestingly, BafA1 further enhanced RSL3-induced LC3-II accumulation and inhibited p62/SQSTM1 degradation in Par-4 overexpressing cells. In conclusion, these findings strongly suggest that Par-4 is crucial in triggering autophagy-mediated ferroptosis. It is important to note that BafA1 also prevented the reduction of GPX4 protein levels induced by RSL3 in both empty vectors and Par-4 overexpressing cells. Based on the new results, we modified our Results section to reflect our point better. Please see the second paragraph on **page 11** for the updated information.

Rebuttal Figure 7: Par-4 plays a crucial role in triggering autophagy-mediated ferroptosis. A172 cells were transiently transfected with either 1 μ g of the empty vector or the Par-4 endogenous human pCMV6-DDK vector. After the transfection, cells were treated with RSL3 2 μ M for 3 h in the presence or absence of 250 nM Bafilomycin A1 (BafA1). Whole-cell lysates were immunoblotted to detect DDK-Par-4, LC3, p62, and GPX4. Actin was used as a loading control.

9. The statement "Induction of 235 ferritinophagy triggers labile iron accumulation, ROS generation, lipid peroxidation, and cell 236 death 7,37,47,48." Is not fully supported by these citations with respect to the cell death part. More accurate would be that induction of ferritinophagy increases the sensitivity to ferroptosis inducers and thereby cell death under conditions of ferroptosis induction.

Response: Thank you for your helpful suggestion. We have now included the appropriate references in the revised manuscript.

The manuscript has been revised to change the statement of new line numbers (263-265) that initially referenced 3,37,47,48 to 44,47,48 and now reads as follows:

44. Zhang Z, *et al.* Activation of ferritinophagy is required for the RNA-binding protein ELAVL1/HuR to regulate ferroptosis in hepatic stellate cells. *Autophagy* **14**, 2083-2103 (2018).
47. Qin X, *et al.* Ferritinophagy is involved in the zinc oxide nanoparticles-induced ferroptosis of vascular endothelial cells. *Autophagy* **17**, 4266-4285 (2021).
48. Zhou H, *et al.* NCOA4-mediated ferritinophagy is involved in ionizing radiation-induced ferroptosis of intestinal epithelial cells. *Redox Biol* **55**, 102413 (2022).

10. As it relates to the data for Par4 controlling NCOA4 / FTH1 degradation: in addition to a western blot, an evaluation of localization is warranted. Specifically is FTH1 no longer trafficking to the lysosome in Par4 KO or KDs?

Response: We appreciate the reviewer for asking this thoughtful question. Our current results provide evidence linking Par-4 to NCOA4-mediated degradation of ferritin heavy chain 1 (FTH1) during ferroptosis. We acknowledge that we currently need to understand this regulation selectivity on the mechanistic level. Therefore, we want to investigate whether up-regulating Par-4 directly activates the ferritinophagy signaling axis, which is being investigated as a separate project in our lab. The results of this follow-up project showed that Par-4 binds to approximately 215 top-ranked

proteins by conducting quantitative mass spectrometry analysis in U87MG cells that overexpressed Par-4. One of the surprising findings was that the 21 kDa FTH1 protein was among the top hits in the Par-4 enforced U87MG cells (**Rebuttal Table 1**), indicating that FTH1 might be a binding partner of Par-4. After reviewing our follow-up data, we have focused on investigating how Par-4 regulates ferroptosis *via* the induction of the ferritinophagy signaling pathway. As the reviewer points out, conducting FTH1 localization on lysosomes in this study will provide further evidence of Par-4's role in controlling NCOA4/FTH1 degradation during ferroptosis. We understand the significance of this experiment and plan to incorporate it into our follow-up project.

Protein IDs	Protein names	Gene names	Mol. weight [kDa]	Score	Intensity
Q961Z0	PRKC apoptosis WT1 regulator protein	PAWR	36.567	47.028	48247000
P02794	Ferritin heavy chain; Ferritin heavy chain, N-terminally processed	FTH1	21.225	11.692	10960000

Rebuttal Table 1. Lists of Par-4 binding proteins identified by mass spectrometry. *Side note: only FTH1 is presented here; data on other Par-4 interacting proteins are excluded.*

11. Figure 4b - the 100 μ M DFO condition alone demonstrates a steady level of FTH1 if not increased. This is inconsistent with the known role of DFO to induce iron starvation and induce FTH1 turnover (decrease in FTH1 band expected). The methods should be updated to indicate the amount of time DFO is applied to the cell. If for a very short time, DFO may not have time to decrease iron levels sufficiently to induce ferritinophagy.

Response: We acknowledge the reviewer's comments that the original manuscript's Fig. 4b shows a constant level of FTH1 in the iron chelator DFO condition. In our revised manuscript, we made some changes. Instead of 100 μ M DFO, we used 50 μ M DFO to pre-treatment A172 cells for 1 hour, followed by RSL3 (2 μ M) for 3 hours. After the treatments, we conducted a western blot analysis to measure the levels of FTH1. We found that RSL3 treatment induced degradation of FTH1 protein level, and this decrease was significantly inhibited by pre-treatment with DFO (**Rebuttal Fig. 8** and **Fig. 4b** in our revised manuscript). Our findings suggest that changes in FTH1 level led to increased availability of the labile iron pool through ferritinophagy. We have included the revised manuscript's updated version of Fig. 4b to replace the previous figure. The figure legends section of our revised manuscript has also been updated accordingly.

Rebuttal Figure 8: A172 cells were pre-treated with DFO (50 μ M) for 1 h, followed by RSL3 (2 μ M) for a further 3 h. Following the treatments, a Western analysis of FTH1 was carried out. The relative density of protein bands was quantified and normalized to the actin of each group, and fold changes were presented in histograms from three independent experiments.

12. Figure 6 - RSL3 is not typically used for IP dosing. Rather RSL3 is used as an intratumoral injection.

Response: Recent studies have shown that RSL3 can inhibit tumor growth in animal models of human cancer when administered intraperitoneally (i.p.) or intratumorally. Specifically, studies have reported successful outcomes using intraperitoneal administration of RSL3 in various mouse models. For example, Yang *et al.* administered RSL3 (5 mg/Kg i.p) biweekly in BALB/c nude mice (*Cell Death Dis.* 2021;12(11):1079), Ghoochani *et al.* administered RSL3 (100 mg/Kg/i.p.) biweekly in NOD-SCID mice (*Cancer Res.* 2021;81(6):1583-1594), and Han *et al.* injected athymic nude mice with RSL3 (50 mg/Kg/i.p.) once every other day for two weeks (*Biochem Biophys Res Commun.* 2021;567:92-98). In addition, recent studies have shown that RSL3 and other GPX4 inhibitor ML162 (40 mg/Kg/i.p./daily) have resulted in tumor regression in BALB/c nude mice (*Cell Metab.* 2023;35(1):84-100.e8). To ensure the validity of our xenograft experiments, we carefully designed animal models based on previous literature [5-8]. Kindly note that for our *in vivo* experiments, we have injected only a very low dose of RSL3 (4.4 mg/Kg/mice, in 100 μ L) IP thrice weekly for 21 days. Furthermore, we conducted H&E staining of vital organs, including the heart, lungs, liver, spleen, and kidney, to evaluate toxicity. The results showed no abnormalities or lesions (refer to **Supplementary Fig. 6b**). Our findings are consistent with *in vitro* observations, indicating that Par-4 plays a crucial role in ferritinophagy-mediated ferroptosis. Based on our results and previous research, we believe intraperitoneal dosing of RSL3 could be a viable option for pre-clinical studies. We have included these additional citations in the revised manuscript to provide evidence and support for our claims (see **Pages 14, 17, and 26**). We hope our interpretation is clear to the reviewer.

References

1. Zhu G, Murshed A, Li H, Ma J, Zhen N, Ding M, Zhu J, Mao S, Tang X, Liu L, Sun F, Jin L, Pan Q. O-GlcNAcylation enhances sensitivity to RSL3-induced ferroptosis via the YAP/TFRC pathway in liver cancer. *Cell Death Discov.* 2021 Apr 16;7(1):83.
2. Yang WS, SriRamaratnam R, Welsch ME, Shimada K, Skouta R, Viswanathan VS, Cheah JH, Clemons PA, Shamji AF, Clish CB, Brown LM, Girotti AW, Cornish VW, Schreiber SL, Stockwell BR. Regulation of ferroptotic cancer cell death by GPX4. *Cell.* 2014 Jan 16;156(1-2):317-331.
3. Gurumurthy S, Goswami A, Vasudevan KM, Rangnekar VM. Phosphorylation of Par-4 by protein kinase A is critical for apoptosis. *Mol Cell Biol.* 2005 Feb;25(3):1146-61.
4. Chakraborty M, Qiu SG, Vasudevan KM, Rangnekar VM. Par-4 drives trafficking and activation of Fas and FasL to induce prostate cancer cell apoptosis and tumor regression. *Cancer Res.* 2001 Oct 1;61(19):7255-63.
5. Yang J, Mo J, Dai J, Ye C, Cen W, Zheng X, Jiang L, Ye L. Cetuximab promotes RSL3-induced ferroptosis by suppressing the Nrf2/HO-1 signalling pathway in KRAS mutant colorectal cancer. *Cell Death Dis.* 2021 Nov 13;12(11):1079.

6. Ghoochani A, Hsu EC, Aslan M, Rice MA, Nguyen HM, Brooks JD, Corey E, Paulmurugan R, Stoyanova T. Ferroptosis Inducers Are a Novel Therapeutic Approach for Advanced Prostate Cancer. *Cancer Res.* 2021 Mar 15;81(6):1583-1594.
7. Han L, Bai L, Fang X, Liu J, Kang R, Zhou D, Tang D, Dai E. SMG9 drives ferroptosis by directly inhibiting GPX4 degradation. *Biochem Biophys Res Commun.* 2021 Aug 27;567:92-98.
8. Yang F, Xiao Y, Ding JH, Jin X, Ma D, Li DQ, Shi JX, Huang W, Wang YP, Jiang YZ, Shao ZM. Ferroptosis heterogeneity in triple-negative breast cancer reveals an innovative immunotherapy combination strategy. *Cell Metab.* 2023 Jan 3;35(1):84-100.e8.

Reviewer #2 (Reviewer Comments to the Author):

In this manuscript, the authors found that prostate apoptosis response-4 (Par-4) promotes ferroptosis possibly through activation of ferritinophagy (autophagic degradation of ferritin) via the nuclear receptor co-activator 4 (NCOA4). The role of NCOA4-mediated ferritinophagy has been clarified in the process of ferroptotic cell death. This study linked Par-4 to NCOA-mediated ferritinophagy in the regulation of ferroptosis. Though the finding expands our knowledge in the understanding of iron metabolism during ferroptosis, there are several issues needed to be addressed before consideration for publication.

Major Points:

1. The rationale to focus the study of Par-4 in the regulation of ferroptosis is not strong enough. Why the authors specifically pick Par-4 as a candidate to start? Because there are so many genes changed during RSL3 treatment, any of them could be a key mediator or regulator of ferroptosis. More evidence should be provided to support the logic. Meanwhile, it's hard to find Par-4 in Fig. 1 A and B. Besides, RSL3 is known to induce autophagy. Is it possible that the induction of Par-4 by RSL3 is simply due to autophagy induction?

Response: Thank you for bringing this to our attention. Although we did discuss some of these points in the original manuscript (Page 4 of the **Introduction** section), we apologize for any lack of clarity that may have caused confusion. We value your concern, as it was crucial in helping us better understand the role of Par-4 in regulating ferroptosis.

Par-4 is a naturally occurring tumor suppressor protein that selectively induces apoptosis in cancer cells while leaving normal, healthy cells unaffected [1]. This specificity towards cancer cells makes Par-4 an attractive target as an anticancer therapeutic. Our lab, like several other different laboratories worldwide, are involved in Par-4-based research. Though Par-4 was initially identified as a pro-apoptotic tumor suppressor, it is clear that Par-4 has a much broader role than inducing apoptosis. Indeed, it contributes to other tumor suppressive pathways, including autophagic cell death, senescence, and metastasis. As a laboratory mainly focusing on elucidating signaling pathways involved in different cell death modalities, we have identified and reported several Par-4-mediated tumor suppressive functions. For instance, we have shown that Par-4 can be cleaved by caspase-3 during apoptosis, which may enhance its pro-apoptotic activity [2,3]. Furthermore, we have demonstrated that Par-4 can potentially trigger other tumor-suppressive mechanisms, such as autophagy and senescence [4,5]. Recent studies have revealed that natural products, dietary supplements, synthetic molecules, and FDA-approved drugs can induce Par-4 secretion to cause apoptosis in primary or metastatic tumors. Based on these findings, one such drug has currently progressed from the bench to clinical trials for anticancer therapy [1].

Our laboratory is currently focused on ferroptosis. Our recent report shows that ferroptosis inhibitors or ROS scavengers can effectively repress the accumulation of cellular ROS and prevent ferroptotic cell death in multiple types of cancer cells [6]. In another recent study, we found that acid sphingomyelinase (an essential enzyme in sphingolipid metabolism)-dependent autophagic degradation of GPX4 plays a crucial role in executing ferroptosis [7]. Our current study involved RNA sequencing analysis, which identified 638 up-regulated and 594 down-regulated gene expressions upon RSL3 treatment. Notably, Par-4 was among the significantly up-regulated genes

(**Fig. 1a and b**). Considering the important role Par-4 may play in different modes of cell death pathways, we decided to focus on this regulation in our further studies. To the best of our knowledge, no research has been done on the effects of Par-4 on ferroptosis and its mechanism of action. In this study, we aim to define the molecular mechanism of ferroptosis, explore the relationship between Par-4 and ferroptosis, and evaluate its potential as a target for cancer treatment.

Meanwhile, it's hard to find Par-4 in Fig.1 A and B

Response: We apologize for not highlighting the PAWR/Par-4 gene in **Fig. 1a and b**. We have revised the manuscript to correct the labeling. The gene of interest, *PAWR* (Par-4), is highlighted by the red arrow in Fig. 1b. Additionally, we want to inform you that the Fig. 1b heatmap has been replaced with a new bar plot, where upregulated and downregulated genes are depicted in red and green color, respectively. This information has now been added to the updated Fig. 1 legend of the revised manuscript.

Besides, RSL3 is known to induce autophagy. Is it possible that the induction of Par-4 by RSL3 is simply due to autophagy induction?

Response: To address the pertinent query of whether RSL3-induced Par-4 is due to autophagy induction, we respectfully request that the reviewer consider the following additional points regarding Par-4's role as a key regulator of ferroptosis established in our study: (1) Our results confirmed that both RSL3- and erastin trigger robust induction of Par-4 in multiple glioblastoma cells, leading to ferroptosis. Indeed, the pre-treatment with ferroptosis inhibitors Fer-1 and Lip-1 prevented cell death. (2) Our functional experiments demonstrated that knockdown of Par-4 (CRISPR, shRNA, and siRNA) almost completely prevented RSL3- and erastin-induced ferroptotic cell death. Importantly, it also affected significant driving events of ferroptosis, such as GPX4 degradation, ROS production, iron accumulation, and lipid peroxidation. Conversely, our findings revealed that overexpression of Par-4 potentiated RSL3 and erastin-induced GPX4 degradation, ROS production, iron accumulation, lipid peroxidation, and cell death. (3) Mechanistically, Par-4 triggers autophagy that degrades ferritin (ferritinophagy) *via* nuclear receptor co-activator 4 (NCOA4), resulting in the release of excessive free iron, enhanced lipid peroxidation, and ultimately, ferroptosis. (4) Inhibition of Par-4 effectively suppressed the NCOA4-mediated ferritinophagy axis. (5) Our new additional data confirmed these results using another GPX4-specific inhibitor, ML210. Genetic inhibition of Par-4 reversed GPX4 degradation, LC3-II conversion, and p62/SQSTM1, FTH1, and NCOA4 in glioma or MEF cells induced by ML210 (Rebuttal Fig. 1a-f and NEW Supplementary Fig. 1a, 2d, 2h, 2l, 3j and 4e). (6) Furthermore, our findings suggest that activation of Par-4 is positively associated with ROS production, which is critical for autophagy-mediated ferroptosis. (7) Finally, it's important to note that attenuation of Par-4 effectively blocked ferroptosis-mediated tumor suppression in mouse xenograft models. We have presented our graphical illustrations here as a rebuttal letter (**Rebuttal Fig. 2**; Fig. 7 in the revised manuscript) for better clarification. Therefore, we argue that our current findings provide substantial evidence to support our hypothesis. We hope the reviewer will agree that these novel results deserve publication in *Communications Biology*.

Rebuttal Figure 2: Schematic representation of Par-4-dependent ROS accumulation plays a critical role in RSL3, ML210-and erastin-induced autophagy-dependent ferroptosis in human glioblastoma cells.

2. Whether another type of ferroptosis inhibitor (iron chelator) could suppress Par-4 expression induced by ferroptosis inducer?

Response: We have conducted new experiments to address the concerns raised by the reviewer. Specifically, we exposed cells with different concentrations of ciclopirox (CPX), a membrane-permeable iron chelator [8-10], for 1 h. This was followed by a 24 h treatment with erastin. The data presented below showed that the CPX pre-treatment did not prevent the up-regulation of Par-4 caused by erastin (**Rebuttal Fig. 3**), suggesting that Par-4 is activated by ferroptosis inducers independent of iron homeostasis. We presented this data in the rebuttal letter, but refrained from including it in the revised manuscript. Notably, the accumulation of labile iron plays a crucial role in ferritinophagy (autophagic degradation of ferritin), leading to lipid peroxidation and ferroptosis [11]. Our results support the hypothesis that the activation of Par-4 triggers the accumulation of labile iron pool and lipid peroxidation, which in turn leads to the induction of autophagy and ferroptosis by degrading ferritin upon exposure to erastin or RSL3.

Rebuttal Figure 3: A172 cells were pre-treated with indicated concentrations of Ciclopirox (CPX) for 1 h. This was followed by treatment with erastin (10 μ M) for 24 h. After the treatment, we conducted a western blot analysis to detect Par-4 levels. Actin was used as a loading control.

3. Fig.1i-k just showed these cells could response for ferroptosis induction, but what's the point to show it here?

Response: In previous studies, Erastin and RSL3 were identified as the first ferroptosis inducers using high-throughput screening of small molecule libraries [12,13]. In our present study, we utilized different ferroptosis inhibitors like ferrostatin-1, liproxstatin-1, and deferoxamine to confirm that erastin- or RSL3-induced cell death in our model cell lines (U87MG and A172 glioma cells) were indeed caused by ferroptosis. It is important to note that none of the other cell death inhibitors, like apoptosis inhibitor Z-VAD-FMK and the necroptosis inhibitors, such as GSK-963, GSK-872, and NSA were able to protect the cell death mediated by erastin and RSL3. Hence, these results confirm that erastin or RSL3 induces ferroptosis, but not other forms of regulated cell death, in U87MG and A172 cells. Part of this discussion has been included in the Results section of the revised manuscript (see **Page 6**). Furthermore, to avoid confusion, Figure 1i-k has been relocated to Supplementary Fig. 1g-i in the revised manuscript.

4. Fig.2a showed that knockdown of Par-4 up-regulated GPX4, which is known to suppress ferroptosis. The author need to prove the role of GPX4 in the Par-4 regulation of ferroptosis.

Response: The reviewer has raised a good point. Additional experiments were conducted to confirm the role of GPX4 in regulating ferroptosis by Par-4 and to determine whether genetic suppression of Par-4 expression affects GPX4 activity. As shown in the **Rebuttal Fig. 4 (NEW Supplementary Fig. 2a** in the revised manuscript), Par-4 suppression significantly increased GPX4 activity in A172 cells in response to erastin. As expected, the knockdown of Par-4 alone had higher GPX4 activity than the control shRNA group. These findings provide additional evidence to support our conclusion. This information has now been added to the updated Supplementary Fig. 2 legend and included in the methods section. In our original manuscript, we proposed a connection between Par-4 and GPX4. We conducted an experiment to determine if inhibiting Par-4 expression would suppress GPX4 degradation. Surprisingly, our results indicate that the genetic deletion of Par-4 through the use of shRNA, CRISPR, or siRNA significantly reduced the degradation of GPX4 induced by both RSL3 and erastin in U87MG and A172 cells (as shown in **Fig. 2a and e** in our revised manuscript). Similar results were obtained using homozygous knockout Par-4^{-/-} primary mouse embryonic fibroblasts (MEFs) (as shown in **Fig. 2f** in our revised manuscript). It is worth noting that the absence or reduction of Par-4 alone was enough to prevent basal GPX4 degradation. On the other hand, overexpression of Par-4 promoted RSL3-induced GPX4 protein degradation in U87MG and A172 cells. Interestingly, overexpression of Par-4 alone was sufficient to cause GPX4 protein degradation in these cells. Altogether, these findings suggest that Par-4 inhibits GPX4 activity by promoting GPX4 protein degradation, which leads to lipid hydroperoxide accumulation and ferroptosis. The figure legends and methods section has been revised accordingly (**Pages 21-22**).

Rebuttal Figure 4a: Par-4 inhibits GPX4 activity during ferroptosis. A172 cells were stably transfected with Par-4-shRNA. After transfection, cells were treated with erastin (10 μ M) for 24 h. Following the treatment, GPX4 activity was measured. Data shown are mean \pm SD; n = 3 samples.

The relevant section of the **Results** has been modified accordingly and now reads as follows:

Par-4 activation is critical for ferroptosis in human glioblastoma cells.

The suppression of Par-4 significantly inhibited both RSL3- and erastin-induced GPX4 protein degradation (Fig. 2a). Subsequently, we determined whether Par-4 knockdown affects the GPX4 activity. The suppression of Par-4 significantly increased GPX4 activity in erastin-treated A172 cells. Interestingly, Par-4 knockdown alone had higher GPX4 activity than the control shRNA group (Supplementary Fig. 2a). Moreover, Par-4 inhibition also prevented labile iron accumulation (Fig. 2b and Supplementary Fig. 2b), and lipid peroxidation (Fig. 2c), which are surrogate markers for ferroptosis. Additionally, suppression of Par-4 significantly prevented both RSL3- and erastin-induced cell death (Fig. 2d).

To further address the reviewer's concern, we tested whether Par-4 regulates the autophagic degradation of GPX4, leading to ferroptosis. We transiently transfected A172 cells with either 1 μ g of the empty vector or Par-4 pCMV6-DDK vector. After the transfection, RSL3 2 μ M was added to the cells for 3 h in the presence or absence of Bafilomycin A1 (BafA1), a widely-used autophagy inhibitor that blocks autophagosome-lysosome fusion. As expected, the data below shows that BafA1 can enhance RSL3-induced LC3-II accumulation and inhibit p62/SQSTM1 degradation in empty vector cells. Surprisingly, BafA1 further enhances RSL3-induced LC3-II accumulation and inhibits p62/SQSTM1 degradation in Par-4 overexpressed cells. It is worth noting that, under the same treatment conditions, BafA1 significantly prevented the reduction of GPX4 protein levels induced by RSL3 in both empty vectors and Par-4 overexpressing cells (see **Rebuttal Fig. 4b and NEW Supplementary Fig. 3m**), proving that Par-4 plays a significant role in the autophagic degradation of GPX4, which ultimately leads to ferroptosis. These findings provide additional evidence to support our conclusion. This new data has now been added to the updated Supplementary Fig. 3 results section of the revised manuscript (see **Page 11**).

Rebuttal Figure 4b: Par-4 regulates the autophagic degradation of GPX4 during ferroptosis. A172 cells were transiently transfected with either 1 μ g of the empty vector or the Par-4 endogenous human pCMV6-DDK vector. After the transfection, cells were treated with RSL3 2 μ M for 3 h in the presence or absence of 250 nM Bafilomycin A1 (BafA1). Whole-cell lysates were immunoblotted to detect DDK-Par-4, LC3, p62, and GPX4. Actin was used as a loading control.

5. Erastin is reported to suppress GPX4 protein level through inhibiting mTORC1 signaling and GPX4 translation. While the authors showed KO of Par-4 restored GPX4 level under erastin treatment. How to explain this if GPX4 proteins translation is already blocked?

Response: We thank the reviewer for the insightful question! A study by Zhang et al. showed that treatment with erastin or cystine deprivation suppresses GPX4 protein levels by inhibiting the Mechanistic Target of Rapamycin Complex 1 (mTORC1) pathway and GPX4 translation, thereby sensitizing cancer cells to ferroptosis [14]. Taking into account of the reviewer's concern, we tested whether an allosteric inhibitor of mTORC1 (rapamycin) could further intensify erastin-induced cell death. On the contrary to the expectation, our result below shows that rapamycin pre-treatment significantly prevented erastin-induced cell death (Rebuttal Fig. 5a), indicating that in our experimental context, erastin is not suppressing GPX4 protein level through inhibiting mTORC1 signaling. This result is further corroborated by our Western blot data, demonstrating that erastin induces both phospho-mTOR and its downstream effector 4EBP1 (rather than decreasing them) (**Rebuttal Fig. 5b**). Of note, under the same treatment conditions, erastin was able to induce GPX4 reduction in A172 cells.

It is important to note that GPX4 reduction during ferroptosis is affected by various factors. In a recent study, we found that acid sphingomyelinase (a sphingolipid metabolic enzyme) facilitates autophagic degradation of GPX4, leading to ferroptosis [7]. Wu and colleagues reported that erastin promotes GPX4 degradation in a CMA (chaperone-mediated autophagy)-dependent manner [15]. GPX4 could also be degraded through selective macroautophagy, and TAX1BP1 (Tax1 binding protein 1) functions as the receptor protein [16]. Furthermore, it was

reported that the ubiquitin-proteasome system is also involved in GPX4 degradation. TRIM46 might be one of the E3 ligases that mediate GPX4 ubiquitination [17]. In this study, we found that Par-4 promotes GPX4 protein degradation *via* autophagy, leading to lipid hydroperoxide accumulation and ferroptosis. We feel that these data go beyond the scope of the current study, and we hope this can stimulate further studies in the future. Therefore, we only show the data in the rebuttal letter and do not include it in the revised manuscript.

Rebuttal Figure 5: A172 cells were treated with erastin (10 μ M) for 24 h in the presence or absence of Rapamycin (5 μ M). (a) Cell viability was measured by using an MTT assay. Data shown are mean \pm SD; n = 3 samples. (b) A172 cells were treated with erastin (10 μ M) for 24 h. Following the treatment, Western blot analysis of the indicated proteins was carried out. Actin was used as a loading control.

6. Free labile iron is known to promote ferroptosis. In fig2g KO of Par-4 increased iron level, which seems contradict to the role of Par-4 in the regulation of ferritinophagy.

Response: We apologize for any confusion caused by Fig. 2g, which may have been due to insufficient information regarding the experimental design. To investigate whether suppressing Par-4 expression impacts free labile iron pool (LIP) accumulation, we used a fluorescent indicator PGSK (Phen GreenTM SK) [18] to evaluate LIP levels. Principally, the labile iron pool interacts with PGSK, resulting in both static and dynamic quenching of the PGSK fluorescence. Based on this, we can determine the concentration of LIP, which will be inversely proportional to the fluorescence generated [19]. In our experiment, we measured the LIP content using fluorescent microscopy and flow cytometry. As shown in **Rebuttal Fig. 6a (Supplementary Fig. 2i** in the revised manuscript), Par-4^{+/+} MEFs treated with RSL3 showed increased LIP, as demonstrated by quenched fluorescence. Surprisingly, homozygous knockout Par-4^{-/-} MEFs did not exhibit significant fluorescence quenching, indicating decreased LIP in response to RSL3 treatment. The PGSK median fluorescence intensity was further verified by flow cytometry (**Rebuttal Fig. 6b** and **Fig. 2g** in the revised manuscript). We obtained similar results when we used shRNA to knockdown Par-4 in U87MG and A172 cells (Fig. 2b in the revised manuscript). We now provided

more detailed information in methods (see pages 23-24 under "Detection of labile iron by imaging and flow cytometry").

To clarify the reviewer's concern straightway, we examined whether suppressing the expression of Par-4 affects the accumulation of free labile iron pool (LIP) using FerroOrange, a Fe²⁺-selective fluorescent probe. The dye was used for fluorescence microscopy to observe the levels of LIP. As shown in **Rebuttal Fig. 6c** and **NEW Supplementary Fig. 2j** in the revised manuscript, RSL3 treatment increased LIP levels in *Par-4^{+/+}* MEFs, as seen by the red-orange fluorescence accumulated in the cells. Knockout *Par-4^{-/-}* MEFs significantly blocked RSL3-induced red-orange fluorescence (decreased LIP) compared to *Par-4^{+/+}* MEFs, as expected. In addition, the FerroOrange median fluorescence intensity (MFI) was confirmed by flow cytometry, as shown in **Rebuttal Fig. 6d** and **NEW Supplementary Fig. 2k**. This finding provides further evidence to support the proposed critical role of Par-4 activation in ferroptosis through ferritinophagy. We have updated the Supplementary Fig. 2 legend panel to include this information and also provided more detailed information in the methods (refer to **Pages 22-23** under "Detection of labile iron by imaging and flow cytometry") of the revised manuscript.

Rebuttal Figure 6: Par-4 promotes iron accumulation during ferroptosis. *Par-4^{+/+}* MEFs and *Par-4^{-/-}* MEFs were treated with RSL3 (2 μ M) for 3 h. After the treatment, the cells were subjected to

PGSK or FerroOrange staining to evaluate intracellular LIP (a and c). Red fluorescence signals were captured and visualized through a fluorescent microscope using constant fluorescence parameters described in the methods section: scale bar, 20 or 50 μm . Subsequently, median fluorescence intensity (MFI) was quantified by flow cytometry analysis. The bar graph shows relative levels of LIP by PGSK or FerroOrange staining in the indicated cells (b and d). Data shown are mean \pm SD; $n = 3$ samples.

7. If Par-4 regulate NCOA4, which KO of Par-4 didn't up-regulate NCOA4 in Fig. 4?

Response: We have presented the original manuscript figures (Fig. 4h-j) as a **Rebuttal Fig. 7** to facilitate our response to this question. To evaluate whether the up-regulation of Par-4 directly contributes to the activation of ferritinophagy in ferroptosis. Genetic ablation of Par-4 by shRNA, CRISPR, or siRNA significantly up-regulated RSL3 or erastin-induced FTH1 and NCOA4 downregulation (highlighted in lane 4 in **Rebuttal Fig. 7a and b**). Similar results were also observed in double knockout Par-4^{-/-} MEFs (highlighted in lane 2 in **Rebuttal Fig. 7c**). We hope this clarification will satisfy the reviewer's concerns.

Rebuttal Figure 7: Par-4 regulates ferroptosis by inducing ferritinophagy activation. (a) U87MG cells were stably transfected with (a) Par-4-shRNA and (b) Par-4 CRISPR. After the transfection, cells were treated with RSL3 (2 μM) for 3 h, and a Western blot analysis of indicated proteins was carried out. (c) Par-4^{+/+} MEFs and Par-4^{-/-} MEFs were treated with RSL3 (2 μM) for 3 h. After the treatment, Western blot analysis of indicated proteins was carried out. Actin was used as a loading control.

8. More evidence are needed to demonstrate the role of NCOA4 in the Par4 regulation of ferroptosis.

Response: We appreciate the reviewer for asking this thoughtful question. Our current results suggest that Par-4 is linked to ferritinophagy mediated by NACO4, which may enhance lipid peroxidation and ferroptosis. We acknowledge that it is important to understand this regulation on the mechanistic level. Therefore, we are investigating whether up-regulating Par-4 directly activates the ferritinophagy signaling axis, which is the focus of work currently being conducted in our lab. The results of these follow-up studies showed that Par-4 binds to approximately 215 top-ranked proteins by conducting quantitative mass spectrometry analysis in U87MG cells that overexpressed Par-4. One of the surprising findings was that the 21 kDa FTH1 protein was among

the top hits in the Par-4 enforced U87MG cells (**Rebuttal Table 1**), indicating that FTH1 might be a binding partner of Par-4. After reviewing our follow-up data, we have focused on investigating how Par-4 regulates ferroptosis *via* the induction of the ferritinophagy (NCOA4/FTH1) signaling pathway. We understand the importance of the question raised by the reviewer and will incorporate it into our follow-up project.

Protein IDs	Protein names	Gene names	Mol. weight [kDa]	Score	Intensity
Q96I20	PRKC apoptosis WT1 regulator protein	PAWR	36.567	47.028	48247000
P02794	Ferritin heavy chain; Ferritin heavy chain, N-terminally processed	FTH1	21.225	11.692	10960000

Rebuttal Table 1. Lists of Par-4 binding proteins identified by mass spectrometry. *Side note: only FTH1 is presented here; data on other Par-4 interacting proteins are excluded.*

9. RSL3 treatment-induced ROS/lipid ROS increased Par-4 and autophagy, how about the regulation of Par-4 by other oxidative stress conditions?

Response: Thank you for providing these insights. Our previous studies have well-documented the redox regulation between oxidative stress and Par-4 [20,21]. Previously, we reported that curcumin-induced Par-4 expression was mediated by generating reactive oxygen species (ROS). In that study, we found that ROS scavengers, such as GSH and NAC, significantly abolished the induction of Par-4 by curcumin [20]. Further, we have shown that extracellular supplementation of H₂O₂ resulted in significant induction of Par-4 expression in both U87MG and U118MG glioma cells. In addition, our previous studies have shown that ROS is involved in the cleavage of Par-4. In both PC3 and DU145 prostate cancer cells, pre-treatment with Sod Pyr, a well-established H₂O₂ scavenger, completely inhibited Par-4-mediated cell death [21]. Our current findings are consistent with our previous research, indicating that in various oxidative stress conditions, Par-4 and ROS molecules mutually regulate each other in a positive feedback manner. We have included part of this discussion in the revised manuscript, which can be found on **Pages 17-18**.

10. In Fig. 6, the authors used RSL3 to induce ferroptosis *in vivo*, which is not appropriate. IKE is the only ferroptosis inducer suitable for *in vivo* study.

Response: We agree with the reviewer's comment that IKE is more suitable for studying ferroptosis in animal studies due to its better potency and metabolic stability [22]. However, several reports indicate that intraperitoneal (i.p) or intratumoral administration of RSL3 can effectively inhibit tumor growth in xenograft animal models of human cancer. For example, (1) ferroptosis was first coined by Stockwell and his colleagues reported that 100 mg/Kg of RSL3 was injected intratumorally into athymic nude mice twice a week for two weeks (*Cell* 2014;156(1-2):317-331). (2) Yang *et al.* administered RSL3 (5 mg/Kg i.p) biweekly in BALB/c nude mice (*Cell Death Dis.* 2021;12(11):1079), (3) Ghoochani *et al.* administered RSL3 (100 mg/Kg/i.p) biweekly in NOD-SCID mice (*Cancer Res.* 2021;81(6):1583-1594), and (4) Han *et al.* injected athymic nude mice with RSL3 (50 mg/Kg/i.p), once every other day for two weeks (*Biochem*

Biophys Res Commun. 2021;567:92-98).). In addition, recent studies have shown that RSL3 and other GPX4 inhibitor ML162 (40 mg/Kg/i.p/daily) have resulted in tumor regression in BALB/c nude mice (*Cell Metab.* 2023;35(1):84-100.e8). To ensure the validity of our xenograft experiments, we carefully designed animal models based on this literature. Kindly note that for our *in vivo* experiments, we have injected only a very low dose of RSL3 (4.4 mg/Kg/mice, in 100 μ L) IP thrice weekly for 21 days. Furthermore, we conducted H&E staining of vital organs, including the heart, lungs, liver, spleen, and kidney, to evaluate toxicity. The results showed no abnormalities or lesions (refer to **Supplementary Fig. 6b**). Based on our results and previous research, we believe that intraperitoneal dosing of RSL3 could be a viable option for pre-clinical studies. We have included these additional citations in the revised manuscript to provide evidence and support for our claims (see **Pages 14, 17, and 26**). We hope our interpretation is clear to the reviewer.

References

1. Cheratta AR, Thayyullathil F, Pallichankandy S, Subburayan K, Alakkal A, Galadari S. Prostate apoptosis response-4 and tumor suppression: it's not just about apoptosis anymore. *Cell Death Dis.* 2021 Jan 7;12(1):47.
2. Thayyullathil F, Pallichankandy S, Rahman A, Kizhakkayil J, Chathoth S, Patel M, Galadari S. Caspase-3 mediated release of SAC domain containing fragment from Par-4 is necessary for the sphingosine-induced apoptosis in Jurkat cells. *J Mol Signal.* 2013 Feb 27;8(1):2.
3. Chaudhry P, Singh M, Parent S, Asselin E. Prostate apoptosis response 4 (Par-4), a novel substrate of caspase-3 during apoptosis activation. *Mol Cell Biol.* 2012 Feb;32(4):826-39.
4. Thayyullathil F, Cheratta AR, Pallichankandy S, Subburayan K, Tariq S, Rangnekar VM, Galadari S. Par-4 regulates autophagic cell death in human cancer cells via up-regulating p53 and BNIP3. *Biochim Biophys Acta Mol Cell Res.* 2020 Jul;1867(7):118692.
5. Subburayan K, Thayyullathil F, Pallichankandy S, Rahman A, Galadari S. Par-4-dependent p53 up-regulation plays a critical role in thymoquinone-induced cellular senescence in human malignant glioma cells. *Cancer Lett.* 2018 Jul 10;426:80-97.
6. Subburayan K, Thayyullathil F, Pallichankandy S, Cheratta AR, Galadari S. Superoxide-mediated ferroptosis in human cancer cells induced by sodium selenite. *Transl Oncol.* 2020 Nov;13(11):100843.
7. Thayyullathil F, Cheratta AR, Alakkal A, Subburayan K, Pallichankandy S, Hannun YA, Galadari S. Acid sphingomyelinase-dependent autophagic degradation of GPX4 is critical for the execution of ferroptosis. *Cell Death Dis.* 2021 Jan 7;12(1):26.
8. Dixon SJ, Lemberg KM, Lamprecht MR, Skouta R, Zaitsev EM, Gleason CE, Patel DN, Bauer AJ, Cantley AM, Yang WS, Morrison B 3rd, Stockwell BR. Ferroptosis: an iron-dependent form of nonapoptotic cell death. *Cell.* 2012 May 25;149(5):1060-72.
9. Stockwell BR, Jiang X. The Chemistry and Biology of Ferroptosis. *Cell Chem Biol.* 2020 Apr 16;27(4):365-375.
10. Aguilera A, Berdun F, Bartoli C, Steelheart C, Alegre M, Bayir H, Tyurina YY, Kagan VE, Salerno G, Pagnussat G, Martin MV. C-ferroptosis is an iron-dependent form of regulated cell death in cyanobacteria. *J Cell Biol.* 2022 Feb 7;221(2):e201911005.
11. Latunde-Dada GO. Ferroptosis: Role of lipid peroxidation, iron and ferritinophagy. *Biochim Biophys Acta Gen Subj.* 2017 Aug;1861(8):1893-1900.

12. Zhu G, Murshed A, Li H, Ma J, Zhen N, Ding M, Zhu J, Mao S, Tang X, Liu L, Sun F, Jin L, Pan Q. O-GlcNAcylation enhances sensitivity to RSL3-induced ferroptosis via the YAP/TFRC pathway in liver cancer. *Cell Death Discov.* 2021 Apr 16;7(1):83.
13. Yang WS, SriRamaratnam R, Welsch ME, Shimada K, Skouta R, Viswanathan VS, Cheah JH, Clemons PA, Shamji AF, Clish CB, Brown LM, Girotti AW, Cornish VW, Schreiber SL, Stockwell BR. Regulation of ferroptotic cancer cell death by GPX4. *Cell.* 2014 Jan 16;156(1-2):317-331.
14. Zhang Y, Swanda RV, Nie L, Liu X, Wang C, Lee H, Lei G, Mao C, Koppula P, Cheng W, Zhang J, Xiao Z, Zhuang L, Fang B, Chen J, Qian SB, Gan B. mTORC1 couples cyst(e)ine availability with GPX4 protein synthesis and ferroptosis regulation. *Nat Commun.* 2021 Mar 11;12(1):1589.
15. Wu Z, Geng Y, Lu X, Shi Y, Wu G, Zhang M, Shan B, Pan H, Yuan J. Chaperone-mediated autophagy is involved in the execution of ferroptosis. *Proc Natl Acad Sci U S A.* 2019 Feb 19;116(8):2996-3005.
16. Xue Q, Yan D, Chen X, Li X, Kang R, Klionsky DJ, Kroemer G, Chen X, Tang D, Liu J. Copper-dependent autophagic degradation of GPX4 drives ferroptosis. *Autophagy.* 2023 Jul;19(7):1982-1996.
17. Zhang J, Qiu Q, Wang H, Chen C, Luo D. TRIM46 contributes to high glucose-induced ferroptosis and cell growth inhibition in human retinal capillary endothelial cells by facilitating GPX4 ubiquitination. *Exp Cell Res.* 2021 Oct 15;407(2):112800.
18. Dixon SJ, Stockwell BR. The role of iron and reactive oxygen species in cell death. *Nat Chem Biol.* 2014 Jan;10(1):9-17.
19. Hirayama T, Nagasawa H. Chemical tools for detecting Fe ions. *J Clin Biochem Nutr.* 2017 Jan;60(1):39-48.
20. Thayyullathil F, Rahman A, Pallichankandy S, Patel M, Galadari S. ROS-dependent prostate apoptosis response-4 (Par-4) up-regulation and ceramide generation are the prime signaling events associated with curcumin-induced autophagic cell death in human malignant glioma. *FEBS Open Bio.* 2014 Aug 30;4:763-76.
21. Rahman A, Pallichankandy S, Thayyullathil F, Galadari S. Critical role of H₂O₂ in mediating sanguinarine-induced apoptosis in prostate cancer cells via facilitating ceramide generation, ERK1/2 phosphorylation, and Par-4 cleavage. *Free Radic Biol Med.* 2019 Apr;134:527-544.
22. Zhang Y, Tan H, Daniels JD, Zandkarimi F, Liu H, Brown LM, Uchida K, O'Connor OA, Stockwell BR. Imidazole Ketone Erastin Induces Ferroptosis and Slows Tumor Growth in a Mouse Lymphoma Model. *Cell Chem Biol.* 2019 May 16;26(5):623-633.e9.
23. Yang J, Mo J, Dai J, Ye C, Cen W, Zheng X, Jiang L, Ye L. Cetuximab promotes RSL3-induced ferroptosis by suppressing the Nrf2/HO-1 signalling pathway in KRAS mutant colorectal cancer. *Cell Death Dis.* 2021 Nov 13;12(11):1079.
24. Ghoochani A, Hsu EC, Aslan M, Rice MA, Nguyen HM, Brooks JD, Corey E, Paulmurugan R, Stoyanova T. Ferroptosis Inducers Are a Novel Therapeutic Approach for Advanced Prostate Cancer. *Cancer Res.* 2021 Mar 15;81(6):1583-1594.
25. Yang F, Xiao Y, Ding JH, Jin X, Ma D, Li DQ, Shi JX, Huang W, Wang YP, Jiang YZ, Shao ZM. Ferroptosis heterogeneity in triple-negative breast cancer reveals an innovative immunotherapy combination strategy. *Cell Metab.* 2023 Jan 3;35(1):84-100.e8.
26. Han L, Bai L, Fang X, Liu J, Kang R, Zhou D, Tang D, Dai E. SMG9 drives ferroptosis by directly inhibiting GPX4 degradation. *Biochem Biophys Res Commun.* 2021 Aug 27;567:92-98.

Reviewer #3 (Reviewer Comments to the Author):

In this study, the authors demonstrated the activation of a multi-faceted tumor-suppressor protein postate apoptosis response-4, Par-4/PAWR in glioblastoma (GBM) U87MG cells treated with ferroptosis activator 1S,3R-Ras SelectiveLethal 3 (RSL3). Functional studies reveal that genetic depletion of Par-4 effectively blocks ferroptosis, whereas Par-4 overexpression sensitizes cells to undergo ferroptosis. The similar experimental works were also conducted in Par-4 wild type and Par-4 knockout primary mouse embryonic fibroblasts (MEFs). Thus, the authors claimed that Par-4, as a tumor suppressor, may play a vital and novel role in ferroptotic cell death upon ferroptosis stimulation. The manuscript seemed to be well documented and experiments also seemed to be well-done.

I have several general suggestions that the authors must have thought of, based on the data presented here. Both of them would make this paper of relevance for a broader audience.

1). Abstract section, the authors claimed that they used an unbiased genome-wide screening to identify Par-4. Actually, the authors provided the RNA sequencing (RNA-seq) analysis. Please keep this consistence.

Response: We thank the reviewer for pointing this out. We have corrected this in our revised manuscript.

2). It has been noticed that Par-4 induces the activation of ferritinophagy via NCOA4. In this case, it is not necessary to confirm the former findings.

Response: We appreciate the reviewer bringing the following concern to our attention. In the previous version of our manuscript, we mentioned in the introduction section (on page 5, line 96) that Par-4 triggers the activation of ferritinophagy via NCOA4. However, we apologize for any confusion caused and would like to clarify that this statement is based on the key findings from our current study. Our research has revealed that the activation of Par-4 induces ferritinophagy via NCOA4. Ferritinophagy is crucial for accumulating the labile iron pool, which generates reactive oxygen species (ROS) and initiates lipid peroxidation, ultimately leading to ferroptosis. We have concluded that Par-4 plays a significant role in the fundamental molecular machinery and signaling pathways of ferroptosis. This is particularly interesting because the regulation of ferritinophagy by Par-4 has not been previously reported.

3). It has been shown that ferroptosis is also an autophagy-dependent cell death. Upon autophagy inducer treatment (such as trehalose or metformin), does Par-4 enhance its tumour suppressor function?

Response: We appreciate this thoughtful remark. A study by Salis et al. reported that the level of Par-4 mRNA expression was increased in response to metformin treatment in MCF-7 breast cancer cells [1], suggesting that metformin increases the Par-4 in the MCF-7 cells, promoting apoptotic activity. This intriguing question urges us to examine whether Par-4 regulates autophagy-mediated

ferroptosis in response to metformin. Our follow-up project will focus on working towards achieving this goal.

4). Page 169-171, "...the expression of p53 over time in U87MG cells (Supplementary Fig. 2j), suggesting that Par-4-mediated ferroptosis induced by RSL3 is independent of p53 expression." This statement may be less conclusive. Since the regulatory role of autophagy by p53 functions diversely on its cellular localization, the author would perform the functional assay of p53, such as the localization change of p53.

Response: We appreciate the reviewer for bringing up this point. Previous research, including studies done in our laboratory, have demonstrated that Par-4 activates p53, leading to tumoricidal function [2,3]. However, on pages 169-171 (as seen in **Supplementary Fig. 2j and now Supplementary Fig. 2o**), it is mentioned that RSL3, the ferroptosis inducer, failed to induce p53 expression over time in U87MG cells. Based on this finding, we have concluded that p53 should be ruled out from this study, and measuring the functional assay is unnecessary, as suggested by the reviewer. We hope that this clarification will satisfy the reviewer's concerns.

References

1. Salis O, Bedir A, Ozdemir T, Okuyucu A, Alacam H. The relationship between anticancer effect of metformin and the transcriptional regulation of certain genes (CHOP, CAV-1, HO-1, SGK-1 and Par-4) on MCF-7 cell line. *Eur Rev Med Pharmacol Sci.* 2014 Jun;18(11):1602-9.
2. Thayyullathil F, Cheratta AR, Pallichankandy S, Subburayan K, Tariq S, Rangnekar VM, Galadari S. Par-4 regulates autophagic cell death in human cancer cells via up-regulating p53 and BNIP3. *Biochim Biophys Acta Mol Cell Res.* 2020 Jul;1867(7):118692.
3. Subburayan K, Thayyullathil F, Pallichankandy S, Rahman A, Galadari S. Par-4-dependent p53 up-regulation plays a critical role in thymoquinone-induced cellular senescence in human malignant glioma cells. *Cancer Lett.* 2018 Jul 10;426:80-97.

Reviewers' comments:

Reviewer #1 (Remarks to the Author):

10. The response to this point is unsatisfactory. FTH1 localization is required to state this. It is a major conclusion from the paper and is a relatively straightforward experiment to perform compared to all the additional data presented with this revision.

"As it relates to the data for Par4 controlling NCOA4 / FTH1 degradation: in addition to a western blot, an evaluation of localization is warranted. Specifically is FTH1 no longer trafficking to the lysosome in Par4 KO or KDs?"

12. Regarding use of RSL3 in vivo. The authors cite many papers that use it in vivo in mice. Just because others use it doesn't mean it works in vivo in mice. Reviewer 2 raises the same point. See this paper:

Randolph et al. Discovery of a Potent Chloroacetamide GPX4 Inhibitor with Bioavailability to Enable Target Engagement in Mice, a Potential Tool Compound for Inducing Ferroptosis In Vivo. *J. Med. Chem.* 2023, 66, 3852–3865

Reviewer #2 (Remarks to the Author):

The reviewer has no more questions regarding the current manuscript. The authors have done a great job on the revision.

Reviewer #3 (Remarks to the Author):

Having looked at the revised manuscript, I believe the authors have done a very good job at providing a more solid and more coherent paper. I am now happy to support publication.

Detailed Point-by-point response to the reviewer's comments

Note to Reviewers: We would like to express our sincere gratitude for taking the time to review our manuscript titled “**Tumor suppressor Par-4 activates autophagy-dependent ferroptosis**” (Manuscript number: COMMSBIO-23-2092A). Your feedback has been incredibly valuable to us, and we appreciate all the effort you have put into providing us with detailed comments. We have carefully considered all your suggestions and have provided a point-by-point response to address them. In this response letter, we have quoted your comments exactly as you wrote them and provided our responses in blue text. We have revised our manuscript and highlighted all the changes in yellow color for your convenience. We believe that your input has greatly improved our manuscript, and we would like to thank you once again for your time and effort.

Reviewer #1 (Remarks to the Author):

10. The response to this point is unsatisfactory. FTH1 localization is required to state this. It is a major conclusion from the paper and is a relatively straightforward experiment to perform compared to all the additional data presented with this revision.

"As it relates to the data for Par4 controlling NCOA4 / FTH1 degradation: in addition to a western blot, an evaluation of localization is warranted. Specifically is FTH1 no longer trafficking to the lysosome in Par4 KO or KDs?"

Response: Thank you for your insightful comments. We appreciate your suggestion to investigate FTH1 localization to lysosomes in Par-4 knockout or knockdown conditions, which could indeed enhance our findings. However, organizing resources for this experiment is time-consuming. We urge the reviewer to consider the following points about Par-4's role in regulating ferroptosis via ferritinophagy: (1) Ferritinophagy, the degradation of ferritin in glioma cells during ferroptosis, is hindered by inhibiting autophagy or lysosomal function, as confirmed by our study using pharmacological (bafilomycin A1) and genetic (Atg-5 and Atg-7) inhibition methods. (2) Upregulation of Par-4 contributes to ferritinophagy during ferroptosis, evidenced by genetic inhibition (shRNA, CRISPR, or siRNA) of Par-4 reversing FTH1 and NCOA4 degradation induced by GPX4 inhibitors (RSL3 and ML210) or Erastin. (3) Par-4 knockout in MEFs mirrored the inhibition of FTH1 and NCOA4 degradation seen in glioma cells. (4) Par-4 overexpression enhanced FTH1 and NCOA4 degradation, promoting ferroptosis. (5) Mass spectrometry analysis of Par-4 overexpressing cells identified FTH1 as a potential binding partner, further supporting Par-4's involvement in NCOA4-mediated ferritin degradation during ferroptosis. These findings establish Par-4's role in ferritinophagy regulation, linking it to ferroptosis. However, we have acknowledged this in our revised manuscript as a limitation of our research. We have also modified our claims in the discussion section to align with our findings. Our ongoing continuation project will cover these crucial experiments.

The relevant section of the Discussion has been modified and now reads as follows (Please refer to Page 17, lines 373-379):

“.....ferritinophagy, a specialized form of autophagy, targets intracellular ferritin for degradation primarily within lysosomes, releasing free iron and inducing ferroptosis^{59,60}. The precise interplay

between Par-4 and ferroptosis, particularly their involvement in ferritinophagy, remains an active area of research. Our results indicate a connection between Par-4 and ferritin degradation via NCOA4 during ferroptosis. However, additional investigations are required to elucidate how Par-4 regulates NCOA4/FTH1 degradation within lysosomes, supplementing our immunoblot confirmation”.

12. Regarding use of RSL3 in vivo. The authors cite many papers that use it in vivo in mice. Just because others use it doesn't mean it works in vivo in mice. Reviewer 2 raises the same point. See this paper:

Randolph et al. Discovery of a Potent Chloroacetamide GPX4 Inhibitor with Bioavailability to Enable Target Engagement in Mice, a Potential Tool Compound for Inducing Ferroptosis In Vivo. *J. Med. Chem.* 2023, 66, 3852–3865.

Response: We extend our appreciation to the reviewer for bringing the Randolph et al. article to our attention. The study highlights the effectiveness of a modified version of RSL3, substituting the chloroacetamide moiety with a sulfone analog. This modification has shown comparable cell-killing activity and enhanced metabolic stability compared to RSL3. Furthermore, the authors propose that RSL3 analogs could be potential single-agent tumor-targeting therapies. The improvements made to address stability issues with RSL3 represent a significant advancement. This enhancement holds promise for enhancing the specificity of GPX4 targeting in tumors, particularly in triggering ferroptosis. Such progress may pave the way for a broader therapeutic window and further exploration in the future.

The reviewer notably recommended a paper that directly aligns with our research. In the initial review, the reviewer noted concerns regarding Figure 6, questioning the typical use of RSL3 for intraperitoneal (IP) dosing. However, the authors cited in the referenced articles did indeed employ RSL3 or its structurally modified version (called 24) for **IP dosing** at doses of 50 and 100 mg/Kg over a 20-day period in their in vivo experiments. This supports our utilization of RSL3 for **IP dosing** at a lower dosage (4.4 mg/Kg/mice) administered thrice weekly over a 21-day duration.

We kindly request the reviewer consider additional details regarding our selection of RSL3 for our animal research. During the preparation phase of our animal study, we referred to reputable studies and contemplated using two ferroptosis activators, IKE and RSL3, for in vivo experiments. While IKE, an erastin analog, boasts higher potency and metabolic stability, reports indicate that RSL3's intraperitoneal (i.p) or intratumoral administration effectively inhibits tumor growth in xenograft animal models of human cancer [*PMID: 34775496*; *PMID: 33483372*; *PMID: 34146907*; *PMID: 36257316*]. IKE primarily targets system Xc⁻ (SLC7A11), whereas RSL3 directly inhibits GPX4's catalytic activity by covalently binding to its selenocysteine residue via its chloroacetamide moiety. Our in vitro findings reveal that Par-4 promotes GPX4 protein degradation, leading to lipid hydroperoxide accumulation and ferroptosis. Hence, we opted for RSL3, effective against GPX4-sensitive tumor models, excluding IKE for our animal studies.

Our in vivo findings align with our in vitro data, highlighting Par-4's importance in ferritinophagy-mediated ferroptosis. Through multiple experiments with RSL3, we consistently observed similar

results, indicating its stability in our animal models. Assessment of toxicity involved H&E staining of vital organs—heart, lungs, liver, spleen, and kidney—revealing no abnormalities or lesions (**Supplementary Fig. 6b**). Moreover, there were no notable changes in animal body weight throughout the study. For clarity, we have included an Excel file titled "**Source Data**," containing the original values of RSL3 treatment (IP dosing) spanning the entire study period (0-21 days), to support Figures 6c and f.

We would like to draw attention to a recent study from this year, demonstrating that administering RSL3 at 50 mg/Kg into tumors triggered tumor regression in BALB/c nude mice (*Cell Death and Disease* 2024 Feb 23;15(2):168; PMID: 38395990). Currently, options for in vivo ferroptosis inducers are limited, with only IKE and RSL3 being viable choices. A reviewer highlighted an article by a pharmaceutical firm in the USA aiming to enhance the stability of RSL3 through structural modifications. These modified versions of RSL3 are anticipated to be commercially available in the future. Consequently, for our future in vivo experiments, we plan to utilize these modified versions of RSL3. To address the reviewer's concern, we've included a citation to this article in the discussion section of our revised manuscript to enhance readers' comprehension. We believe this clarification effectively addresses the reviewer's query.

The manuscript has been updated under the discussion section (please refer to Page 17, lines 380-381 for the revised text) and now reads as follows:

Several reports suggest that RSL3 or its structurally modified analog, administered intraperitoneally (i.p), was used in vivo for xenograft models⁵²⁻⁵⁶.

Reviewer #2 (Remarks to the Author):

The reviewer has no more questions regarding the current manuscript. The authors have done a great job on the revision.

Response: We thank the reviewer for the kind support.

Reviewer #3 (Remarks to the Author):

Having looked at the revised manuscript, I believe the authors have done a very good job at providing a more solid and more coherent paper. I am now happy to support publication.

Response: We thank the reviewer for the kind support.

REVIEWERS' COMMENTS:

Reviewer #1 (Remarks to the Author):

The authors have added statements to their manuscript that adequately state the limitations.

Detailed Point-by-point response to the reviewer's comments

Reviewer #1 (Remarks to the Author):

The authors have added statements to their manuscript that adequately state the limitations.

Response: We thank the reviewer for the kind support.